# Chemoproteogenomic stratification of the missense variant cysteinome

Heta Desai [1,2], Katrina H. Andrews[1], Kristina V. Bergersen[3], Samuel Ofori[1], Fengchao Yu [4], Flowreen Shikwana[1,5], Mark A. Arbing[1,6], Lisa M. Boatner [1,5], Miranda Villanueva [1,2], Nicholas Ung[1], Elaine F. Reed[3], Alexey I. Nesvizhskii [4,7] & Keriann M. Backus [1,2,5,6,8,9] ✉

Cancer genomes are rife with genetic variants; one key outcome of this variation is widespread gain-of-cysteine mutations. These acquired cysteines can be both driver mutations and sites targeted by precision therapies. However, despite their ubiquity, nearly all acquired cysteines remain unidentified via chemoproteomics; identification is a critical step to enable functional analysis, including assessment of potential druggability and susceptibility to oxidation. Here, we pair cysteine chemoproteomics—a technique that enables proteome-wide pinpointing of functional, redox sensitive, and potentially druggable residues—with genomics to reveal the hidden landscape of cysteine genetic variation. Our chemoproteogenomics platform integrates chemoproteomic, whole exome, and RNA-seq data, with a customized two-stage false discovery rate (FDR) error controlled proteomic search, which is further enhanced with a user-friendly FragPipe interface. Chemoproteogenomics analysis reveals that cysteine acquisition is a ubiquitous feature of both healthy and cancer genomes that is further elevated in the context of decreased DNA repair. Reference cysteines proximal to missense variants are also found to be pervasive, supporting heretofore untapped opportunities for variant-specific chemical probe development campaigns. As chemoproteogenomics is further distinguished by sample-matched combinatorial variant databases and is compatible with redox proteomics and small molecule screening, we expect widespread utility in guiding proteoform-specific biology and therapeutic discovery.

The average human genome differs from the reference at roughly 5 million sites (~0.1% of the genome)[1]. This profound genetic variation gives rise to human diversity and disease. While protein-altering single nucleotide variants (SNVs) make up a small fraction of all known variants, most known disease-causing mutations are found in protein-coding sequences. Thus, understanding whether a genetic variant is translated and deciphering the impact of that variant on protein activity are critical steps for

[1]Biological Chemistry Department, David Geffen School of Medicine, UCLA, Los Angeles, CA, USA. [2]Molecular Biology Institute, UCLA, Los Angeles, CA, USA. [3]Department of Pathology and Laboratory Medicine, David Geffen School of Medicine, UCLA, Los Angeles, CA, USA. [4]Department of Pathology, University of Michigan, Ann Arbor, MI, USA. [5]Department of Chemistry and Biochemistry, UCLA, Los Angeles, CA, USA. [6]UCLA-DOE Institute for Genomics and Proteomics, UCLA, Los Angeles, CA, USA. [7]Department of Computational Medicine and Bioinformatics, University of Michigan, Ann Arbor, MI, USA. [8]Eli and Edythe Broad Center of Regenerative Medicine and Stem Cell Research, UCLA, Los Angeles, CA, USA. [9]Jonsson Comprehensive Cancer Center, UCLA, Los Angeles, CA, USA. ✉e-mail: kbackus@mednet.ucla.edu

characterizing the functional and therapeutic relevance of genomic variation.

Proteogenomic studies that implement custom variant-containing sequence databases have made significant inroads into the former challenge, enabling proteome-wide detection of protein-coding variants, including single amino acid variants (SAAVs) and splice variants[2–8] and have even achieved enhanced variant coverage when paired with ultra-deep fractionation[9,10]. These studies all share the same general data processing pipelines. Variant calling is performed on next-generation sequencing (NGS) data. Customized databases featuring both canonical protein sequences and sequences encoding SAAV-, insertion/deletions (indels)-, or splice variant-proteins are then generated using customized tools such as Spritz[11], CustomProDB[12], Galaxy-P[13], and sapFinder[14]. There are two central complexities to these pipelines that have only recently begun to be addressed. The first challenge is that, by relying on exome-only sequencing and short-read sequencing, the relative proximity of two or more variants in the same gene (whether they are on the same or opposite chromosomes) is not typically apparent. A notable exception is the recent integration of long-read sequencing for de-novo database construction with sample-specific proteomics to characterize protein isoforms[15]. However, such search strategies also introduce higher chances of false positive identification[16].

One solution to the false discovery rate (FDR) challenge is to calculate a class-specific FDR (separating the FDR calculations for the variant-containing peptides and reference peptides)[16]. A strategy for assessing class-specific FDR is a two-stage FDR database search[17]. While the implementation of such strategies in prior proteogenomic studies highlights the importance of rigorous statistical validation of identified variant-containing peptides[17–19], the requirement for customized pipelines has so far limited widespread adoption. Together with these valuable technical innovations that enable rigorous proteogenomic identification of SAAVs, an additional key opportunity for proteogenomics is the delineation and monitoring of functional and disease-associated variants.

Mass spectrometry-based chemoproteomics assays are ideally suited to shed light on genetic variant significance. Exemplifying this utility, chemoproteomics methods have been established that measure amino acid intrinsic reactivity, which is indicative of functionality, potential druggability, and sensitivity to post-translational modifications[20–28]. However, SAAVs are almost universally missed by chemoproteomic studies. The key reason for this gap is that most genetic variants are not found in reference protein sequence databases used to identify peptides from acquired tandem mass spectrometry (MS/MS) data[20,21,29–36].

Across all chemoproteomic-detectable residues, cysteine is uniquely suited to proteogenomic analysis. Quite surprisingly given the relative rarity of cysteine (2.3% of all residues in a human reference proteome)[37], cysteine is the most commonly acquired amino acid due to somatic mutations in human cancers[38], with net 5% gained and 1% lost cysteines encoded by the 2 million coding mutations that have been identified in human cancers (Catalog of Somatic Mutations [COSMIC] database). Given the unique chemistry of the cysteine thiol, including its nucleophilicity and sensitivity to oxidative stress, a subset of these residues almost unquestionably has a substantial impact on protein function. Exemplifying this paradigm, a number of driver mutations are gained cysteines, including KRAS G12C, SHP2 Y279C, FGFR S249C, and IDH1 R132C[39–43]. A likely reason for the ubiquity of cysteine acquisition is the comparative instability of CpG motifs; C-T transitions are nearly ten times more common than other missense mutations in cancer[44], and these transitions should favor gain-of-cysteine codons.

Here, we develop and deploy chemoproteogenomics as an integrated platform tailored to capture the missense variant cysteinome. Chemoproteogenomics unites a missense-variant-focused proteogenomic pipeline with mass spectrometry-based cysteine chemoproteomics. By mining publicly available datasets, including COSMIC, dbSNP, and ClinVar, we reveal that gain-of-cysteine variants are a ubiquitous consequence of genetic variation. We further reveal that DNA repair-deficient cell lines are particularly enriched for acquired cysteines, together with a general high burden of rare and predicted deleterious variants. Guided by these discoveries, we generate combinatorial cell-specific custom databases built from whole exome and RNA-Seq data for eleven cell lines. Chemoproteogenomic analysis with a user-friendly FragPipe computational platform, extended to support two-stage database search and FDR estimation, identified >1400 total unique variants, including 677 chemoproteomic enriched variant-proximal cysteines and 104 gain-of-cysteines. Chemoproteogenomics also robustly identifies ligandable SAAVs that alter cysteine oxidation state and outperforms bulk proteogenomic analysis for capture of SAAVs with lower variant allele frequency. The utility of chemoproteogenomics is further showcased through our identification of ligandable genetic variants that alter cytokine activity for HMGB1 and protein interactions for CAND1. In sum, chemoproteogenomics sets the stage for an enhanced global understanding of the functional and therapeutic relevance of the missense variant proteome.

## Results

### Variant peptide identification enabled by MSFragger two-stage database search and false discovery rate (FDR) estimation

To enable chemoproteogenomic identification of SAAV-containing peptides, we established a customized proteogenomics pipeline (Fig. 1A). Motivated by a prior report[3] that demonstrated proteogenomic sample searches performed with sample-specific databases both improved coverage (~45% more variants) and decreased rates of SAAV peptide false discovery, we generated a cell line-specific variant peptide database from HEK293T RNA-seq data (Fig. 1A, Supplementary Fig. 1 and Supplementary Data 1). Next, to afford a reduction in the likelihood that a variant peptide will be mismatched to wild-type spectra[17], we established a two-stage database search and FDR control scheme (Fig. 1A), using an MSFragger[45,46] command line pipeline within FragPipe computational platform. In this strategy, the first search of acquired MS/MS spectra is performed against a reference database of canonical protein sequences. Subsequently, peptide-to-spectrum (PSM) matches identified with a certain high level of confidence (e.g., passing 1% FDR) are removed, and the remaining spectra are then searched against a variant-containing, sample-specific database.

We then subjected our chemoproteogenomics pipeline to benchmarking by generating a set of high-coverage cysteine chemoproteomics datasets (Fig. 1B) in which reference and variant proteinacious cysteines in cell lysates labeled with iodoacetamide alkyne (IAA)[20] and conjugated isotopically labeled 'light' ($^1H_6$) or heavy' ($^2H_6$) biotin-azide reagents[47] (+ 6 Da mass difference between the reagents) were combined pairwise in biological triplicate at different H/L ratios (1:1,10:10, 1:4, 4:1, 1:10, and 10:1). By searching these datasets using our two-stage FDR search, we sought to validate the accuracy of variant identification. Peptide quantification using IonQuant[48,49], following the workflow shown in Fig. 1A, revealed MS1 intensity ratios for both canonical and variant peptide sequences that matched closely with the expected values (Fig. 1C and Supplementary Data 1). We also compared the retention times of the heavy- and light-peptides and observed a ~2-3 sec shift for the deuterated heavy sequences for both the variant and canonical peptide sequences (Fig. 1D and Supplementary Data 1). These retention time shifts are consistent with our previous study[47] and with prior reports[50,51]. Analogous to studies that utilize isotopically enriched synthetic peptide standards to validate peptide sequences[52–54], the observed co-elution of both heavy and light variant peptides provides further evidence to support the low FDR of our data processing pipeline. Lastly, the high concordance between observed

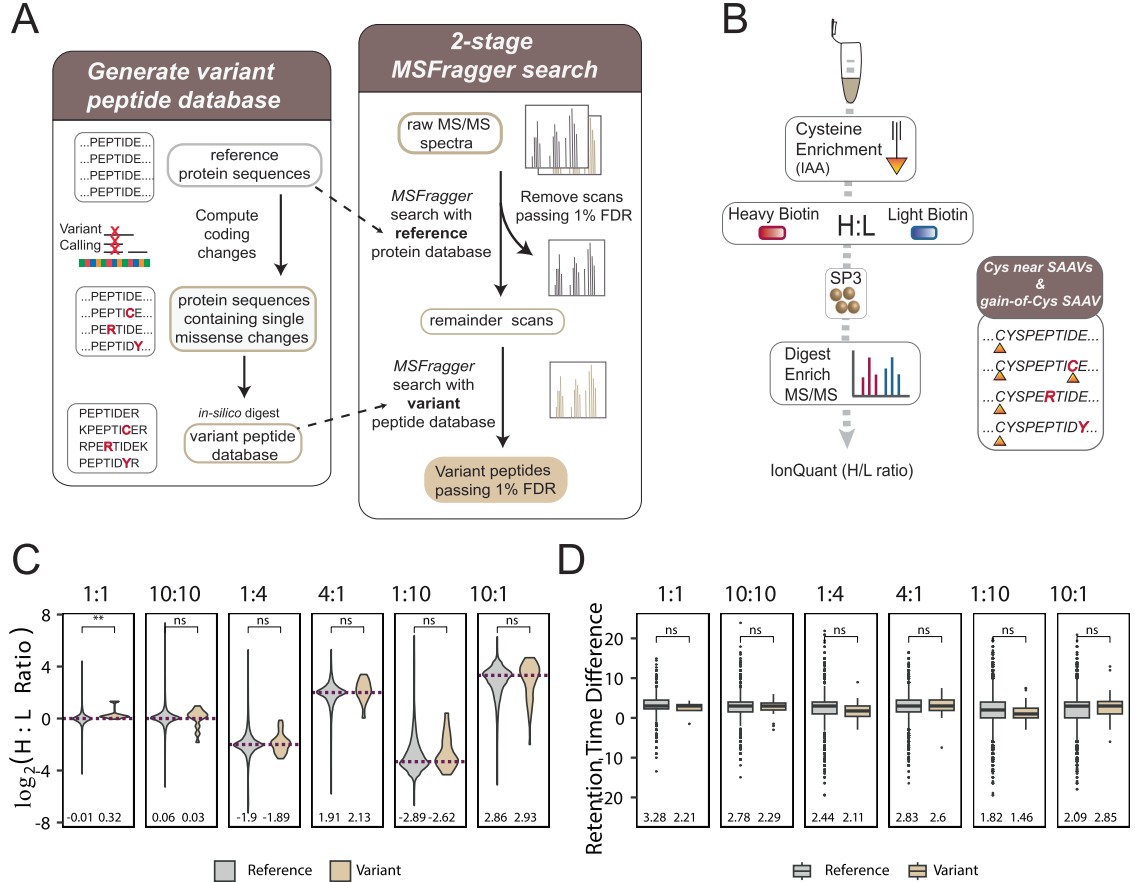

**Fig. 1 | Establishing an MSFragger-search pipeline for variant peptide identification. A** Two-stage FDR MSFragger-enabled variant searches–variant databases are generated from non-redundant reference protein sequences that are *in-silico* mutated to incorporate sequencing-derived missense variants followed by two-stage FDR MSFragger/PeptideProphet search to identify confident variant-containing peptides. First, raw spectra are searched against a normal reference protein database, confidently matched spectra (passing 1% FDR) are removed, and the remainder of spectra are searched with a variant tryptic database.
**B** Chemoproteomics workflow to validate heavy and light biotin[47]. HEK293T cell lysates were labeled with pan-reactive iodoacetamide alkyne (IAA) followed by 'click' conjugation onto heavy or light biotin azide enrichment handles in known ratios. Following neutravidin enrichment, samples are digested and subjected to

MS/MS analysis. **C** Heavy to light ratios (H:L) from triplicate datasets ($n = 3$) comparing identifications from reference and variant searches; mean ratio value indicated, *dashed lines* indicate ground-truth $\log_2$ ratio, statistical significance was calculated using a two-sided Mann-Whitney U test, **$p < 0.01$, ns $p > 0.05$ (1:1, $p = 0.002$; 10:10, $p = 0.083$; 1:4, $p = 0.84$, 4:1, $p = 0.093$; 1:10, $p = 0.056$; 10:1. $p = 0.061$). **D** Retention time difference for heavy and light identified peptides for reference and variant searches; mean value indicated, statistical significance was calculated using a two-sided Mann-Whitney U test, ns $p > 0.05$ 05 (1:1, $p = 0.47$; 10:10, $p = 0.42$; 1:4, $p = 0.45$, 4:1, $p = 0.57$; 1:10, $p = 0.13$; 10:1. $p = 0.34$)… Box plot center line, median; limits are upper and lower quartiles; 1.5x interquartile range. Proteomic data is found in Supplementary Data 1 and source data in the Source Data file.

and expected MS1 ratios provides compelling support for the use of the heavy and light biotin azide reagents in competitive cysteine-reactive compound screens, in which elevated MS1 intensity ratios are indicative of a compound-modified cysteine.

## FragPipe graphical user interface (GUI) with improved two-stage MSFragger search and FDR estimation

Motivated by the multi-faceted uses of the two-stage FDR search pipeline for general proteogenomic applications, we next simplified the search workflow by establishing a semi-automated execution of these searches in FragPipe (see Supplementary Discussion for details). To further improve the sensitivity of variant peptide identification, we added an option to run MSBooster and Percolator instead of PeptideProphet (Supplementary Fig. 2). As part of our semi-automated search pipeline, we enabled compatibility with isobaric labeling reagents, which we expect will further broaden the utility of our approach (Supplementary Fig. 3). Using the GUI features, we observed comparable coverage for both the command-line and automated GUI implementations of the two-stage FDR search with a slight increase in numbers of identifications observed for

datasets processed with MSBooster and Percolator (Supplementary Fig. 4 and Supplementary Data 1). The ratio differences between variant and reference cysteine peptides are comparable (Supplementary Fig. 2). In total we identified 50 missense variants at the protein level, including 11 acquired cysteines and 39 proximal to reference cysteines. This very low coverage of variant data prompted us to reconsider our cell line selection and to prioritize the addition of whole exome sequencing data to enhance the coverage of functionally significant variants.

## High missense burden cancer cell lines are rich in acquired cysteines, including in census genes

We hypothesized that the genomes of missense-variant rich cell lines would similarly encode a high burden of acquired cysteine SAAVs and SAAVs proximal to reference cysteines and, therefore, could serve as useful model systems for establishing cysteine chemoproteogenomics. To test this hypothesis and establish a useful toolbox of cell lines and genomics data for our chemoproteogenomics platform, we analyzed the variant burden across all cell line data available in the Catalog of Somatic Mutations in Cancer Cell Lines Project

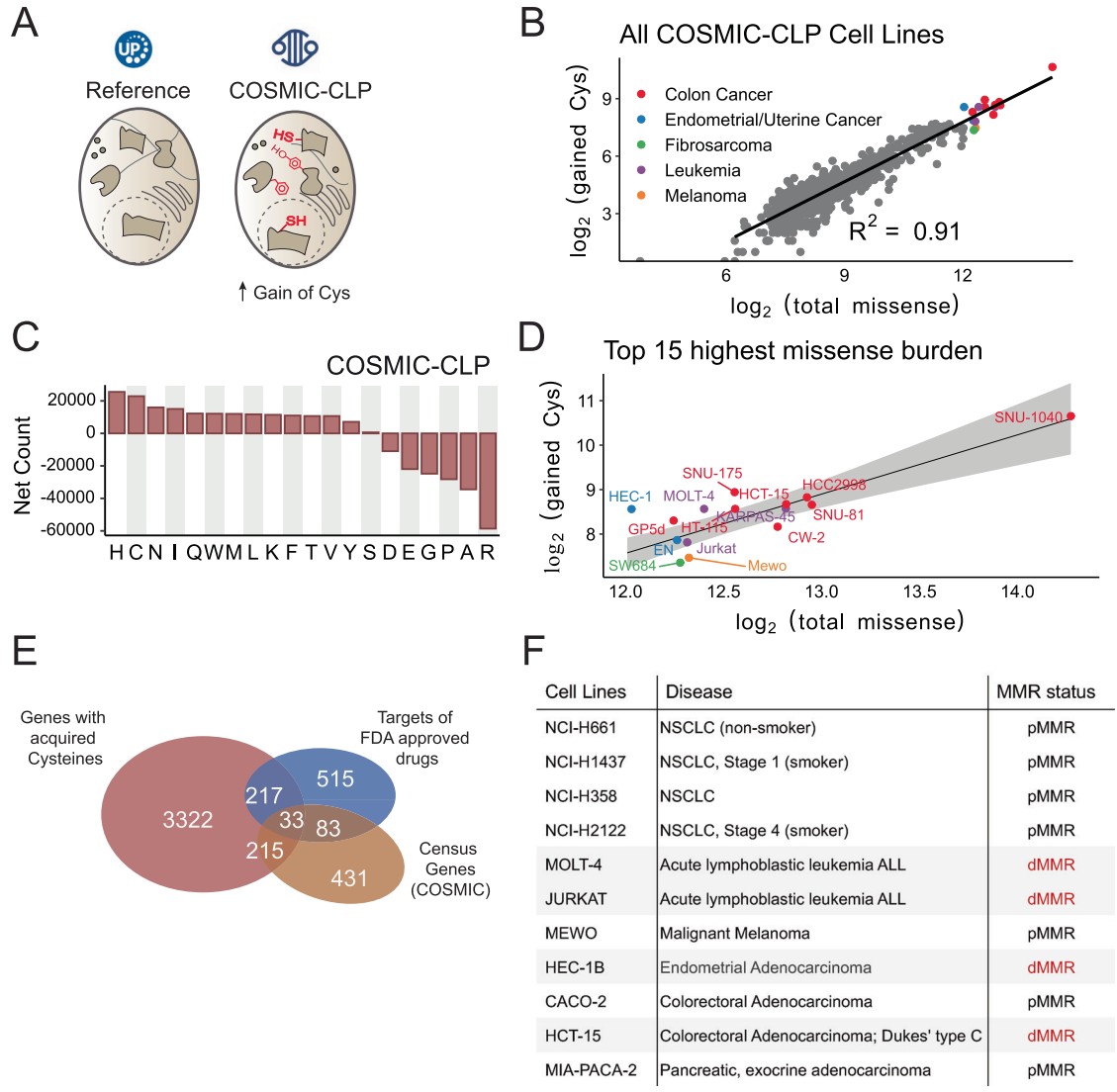

Fig. 2 | **Acquired cysteines are prevalent across cancer genomes, particularly for high missense burden cell lines. A** The full scope of acquired cysteines in the COSMIC Cell Lines Project (COSMIC-CLP, cancer.sanger.ac.uk/cell_lines) (v96)[55,56] were analyzed. **B** 1020 cell lines stratified by the number of gained cysteines and total missense mutations; color indicates cancer type for the top 15 highest missense count cell lines. **C** Net missense mutations (gained-lost) from COSMIC-CLP (v96). **D** Top 15 cell lines with highest missense burden from panel (**B**); linear regression and 95% confidence interval shaded in gray. **E** Overlap of genes with acquired cysteines in top 15 subsets from panel (**B**) with Census genes and targets of FDA-approved drugs. **F** Panel of cell lines used in this study with MMR status (dMMR = deficient mismatch repair, pMMR=proficient mismatch repair). Data is found in Supplementary Data 2 and source data is in the Source Data file.

database(COSMIC-CLP)[55,56] (Fig. 2A). A comparatively small subset of cell lines was observed to be particularly missense rich, with only 15 out of 1020 total cell lines in COSMIC harboring 77,693 or ~18% of the ~2 million unique missense variants cataloged (Fig. 2B, Supplementary Fig. 5A and Supplementary Data 2). Gratifyingly and consistent with our hypothesis, cysteine was a top-gained amino acid, both across all COSMIC cell line variants (23,220 total acquired cysteines; 5.4% of total COSMIC cell line mutations) and across the top 15 high missense burden cell lines (4725 total acquired cysteines found in 3, 688 genes), with a strong correlation between overall missense burden and net acquired cysteines (Fig. 2B–D and Supplementary Fig. 6). These data suggested that a comparatively small fraction of cell lines could prove useful for proteogenomic analysis of somatic variants in cancer.

Nearly 30% (219/738) of the Census genes (v98) identified in the top 15 missense-rich cell lines were found to harbor one or more gained cysteines (Supplementary Data 2), and <10% of these genes have been targeted by FDA approved drugs[29,57] (Fig. 2E).

## dMMR cell lines are enriched for SAAVs, including acquired cysteines

Microsatellite instability (MSI) caused by deficiencies in mismatch repair (dMMR), as opposed to functional MMR or proficient mismatch repair (pMMR), is a prominent feature of missense mutation-rich cell lines. Notably, 7 of the top 15 missense cell lines in COSMIC are known mismatch repair deficient cell lines[58–60] (Fig. 2D and Supplementary Fig. 7), and only MeWo cells, which are derived from metastasized melanoma, were reported to be microsatellite stable (MSS)[58]. The majority of missense-rich cell lines, including the dMMR lines were observed to encode between 5000 and 10,000 total SAAVs and 200 and 500 acquired cysteine SAAVs (Supplementary Fig. 5). By causing $C \rightarrow T$ mutations primarily at CpG sites, the mutational signature of defective mismatch repair (SBS6) should favor gain-of-cysteine[61]. While the missense-rich nature of the dMMR cell lines provides an exciting opportunity for high variant coverage proteogenomics, the predominance of MSI across the cell line panel together with the marked overrepresentation of colorectal carcinoma (CRC) cell lines

(Fig. 2B and Supplementary Fig. 7) prompted us to broaden our cell line panel to better represent genetic variation and to further assess how cell line MSI status impacted variant content.

## An expanded cell line panel incorporates high-value acquired cysteines

Given the considerable interest in targeting G12C KRAS, we opted to add several *KRAS* mutated cell lines to our panel (MIA-PACA-2, H2122, and H358) in order to favor the detection of the G12C peptide. Notably, the smoking-associated mutational signature is C → A/G → T[62], which should also favor gain-of-cysteines. Therefore, we additionally sought to test whether smoking-associated NSCLC-derived H2122 and H1437 adenocarcinoma cell lines would be enriched for acquired cysteines when compared to other pMMR cell lines, including lung cancer cell lines (H358 NSCLC and H661 metastatic large cell undifferentiated carcinoma (LCUC) lung cancer cell lines). Lastly, we opted to include CACO-2 cells, an MSS CRC cell line, given the preponderance of

missense rich dMMR CRC cell lines. Our prioritized cell line panel features 11 cell lines in total (2 female and 9 male) spanning 6 tumor types and encoding 22,559 somatic variants and 1296 somatic acquired cysteines, as annotated by COSMIC-CLP (Fig. 2F and Supplementary Data 2), with aggregate enrichment for gained cysteines observed for the entire panel (Supplementary Figs. 8, 9). Of the proteins that harbor gained cysteines, 486 are Census genes, and 5% are targeted by FDA-approved drugs (Supplementary Data 2).

## Incorporating rare variants into our proteogenomic pipeline

COSMIC and related cancer databases often do not report germline variants found in the general population. Therefore, to enable chemoproteogenomic assessment of non-cancer-associated SAAVs, we sequenced exomes and RNA of our cell lines and subjected NGS reads to variant-calling (Fig. 3A and Supplementary Fig. 10). For all 11 cell lines sequenced, we identified average 82% of the variants reported in COSMIC-CLP, including known driver mutations, and 70% of missense

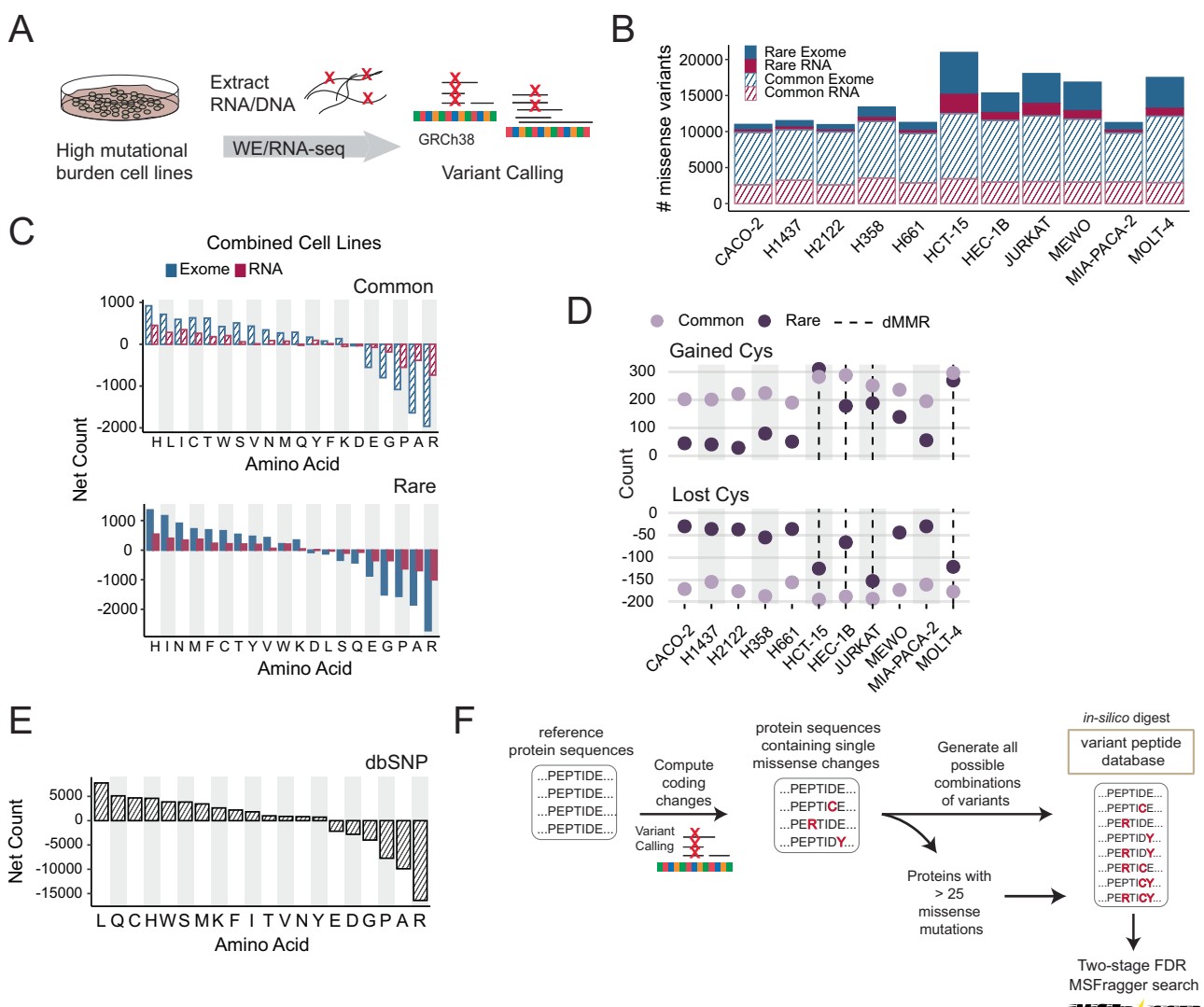

**Fig. 3 | Incorporating variants into sample-specific search databases.**
**A** Sequencing portion of the 'chemoproteogenomic' workflow to identify chemo-proteomic detected variants–extracted genomic DNA or RNA from cell lines undergo sequencing followed by variant calling using Platypus (v0.8.1)[118] and GATK-Haplotype Caller (v4.1.8.1)[119] for RNA and exomes respectively and predicted missense changes were computed. **B** Total numbers of missense mutations identified from either RNA-seq or WE-seq; stripe vs solid denotes common and rare variants. **C** Net amino acid changes for all cell lines combined. **D** Totals of gained and lost

cysteine in each cell line separated by rare and common variants, dashed line indicates dMMR cell lines. **E** Net missense mutations (gained-lost) from dbSNP (4-23-18)[65]. **F** Non-synonymous changes are incorporated into reference protein sequences, and combinations of variants are generated for proteins with less than 25 variant sites to make customized FASTA databases. Details in methods. Supplementary Data 3 and Supplementary Data 4 and source data in the Source Data file.

mutations reported by Cancer Cell Line Encyclopedia (CCLE)[58] databases (Supplementary Data 3). 9485 total rare variants and 22,010 total common variants were identified that had been not previously reported in COSMIC-CLP. Of those variants not in the COSMIC-CLP 237 are annotated as pathogenic/likely pathogenic/VUS in ClinVar, and 1251 variants encode acquired cysteines (Supplementary Data 3). Analysis of DNA damage repair-associated genes revealed specific mutations (Supplementary Data 3), including *DDB2* R313* in MeWo cells, which provide an explanation for the previously unreported high missense burden—inactivating mutations in DDB2 are implicated in deficient nucleotide excision repair[63]. Pointing towards opportunities to improve coverage of reference-cysteine-containing peptides, 16,381 total reference cysteines were located proximal (within 10AA) to missense variants, including 10,508 variants not previously identified in the COSMIC-CLP (Supplementary Data 3).

We also compared the variant landscape of each cell line with the goal of identifying ubiquitous common variants together with rare and cell line-specific variants. As with our analysis of COSMIC-CLP, we detected a high missense burden for the dMMR cell lines compared to the pMMR cell lines (Fig. 3B). In total, 1634 variants were shared across all cell lines, and 34,636 were unique to individual cell lines, which illustrates the added value of analyzing multiple cell lines. Notably, when compared to the pMMR cell lines, we found that nearly all of the dMMR cell lines, most notably HCT-15 and Molt-4 cell lines, were comparatively enriched for rare variants and particularly rare, acquired cysteines compared to the pMMR cysteines (Fig. 3B,C), irrespective of sequencing coverage (Supplementary Fig. 11). In contrast, both pMMR and dMMR genomes harbored comparable numbers of common variants, including common acquired and lost cysteines (Fig. 3D and Supplementary Figs. 12–14). This finding points towards an opportunity to use dMMR cell lines for proteogenomic analysis of rare variants and particularly rare acquired cysteines.

Looking beyond cysteine acquisition, we also considered how the broader missense amino acid signature varied across cell lines to identify other features that might impact our proteogenomic pipeline. For common variants, the amino acid gain/loss signatures were generally consistent across cell lines (Fig. 3D), including for smoking versus non-smoking-associated lung cancer cell lines (Supplementary Fig. 15), characterized by marked enrichment for acquired histidine and cysteine together with loss-of-arginine (Supplementary Figs. 12–14). For rare variants, cell-line-specific differences in SAAV content were observed, most notably when comparing the dMMR to pMMR cell lines (Fig. 3D and Supplementary Figs. 12–14). MeWo cells harbored many gains of rare phenylalanine and lysine (Supplementary Fig. 13), consistent with UV radiation-induced pyrimidine dimers (Supplementary Fig. 16 and Supplementary Data 3). Thus, we expect that the ubiquity of loss-of-arginine together with the MeWo gain-of-lysine signature should alter the tryptic peptide landscape, and proteogenomic analysis should enable improved detection of this class of missense variants.

### Acquired cysteines are ubiquitous in both healthy and diseased genomes

Looking beyond cancer variants, we were also interested in determining whether our chemoproteogenomic platform could prove useful for the study of acquired cysteines more broadly, including ubiquitous common variants and rare variants that may have links to monogenic disorders. We hypothesized that gain-of-cysteine missense variants should also be ubiquitous in healthy genomes, due to the comparative instability of CpG—a key consequence of this instability is the frequent loss-of-arginine codons (4/6 CG dinucleotides)[64]. To test this hypothesis, we aggregated and quantified the amino acid changes resulting from common missense variants reported by dbSNP[65] (4-23-18), a repository of single nucleotide polymorphisms, and ClinVar[66] (09-03-22), a repository of variants with reported pathogenicity. We

find that cysteine acquisition is the third most common consequence of missense variants identified in dbSNP (Fig. 3E and Supplementary Data 2) for common variants—common variants are defined by NCBI as of germline origin and/or with a minor allele frequency (MAF) of ≥ 0.01 in at least one major population, with at least two unrelated individuals having the minor allele. Analogous stratification of variants reported by ClinVar also revealed a preponderance of gained cysteines compared with lost cysteines, albeit to a more modest degree than that observed for cancer genomes (Supplementary Fig. 17 and Supplementary Data 2). For the pathogenic variant subset of ClinVar, both gain- and loss-of-cysteine and gain-of-proline were frequently observed (Supplementary Fig. 17). Comparing the variants in our cell line panel to those found in dbSNP and ClinVar, we find that 25,735 (dbSNP/common) and 3982 (ClinVar; 3409 common and 573 rare) variants are found in our cell line panel, which highlights additional opportunities for analysis of acquired cysteines relevant to other genetic contexts, including rare disease and healthy genomes (Supplementary Data 3). Notably, 3560 variants are found within 5 amino acids of additional variants. The proximity of missense variants, particularly rare and common variants, points toward a need for combinatorial databases[67] for proteogenomics.

### Deploying chemoproteomics with combinatorial databases improves coverage of acquired cysteines and proximal variants

To establish our proteogenomics pipeline, we were inspired by the recent report[68] combinatorial databases to improve the detection of proximal SAAVs, such as the aforementioned variants that are found within 30 amino acids. To improve the detection of such variants, we established an algorithm (Supplementary Fig. 1B) to generate all combinations of SAAVs derived from both RNA/WE-seq data within 30 amino acids flanking the variant site. These combinations were then converted into a peptide FASTA database containing two tryptic sites flanking each variant site (Fig. 3F). On average, >4500 total multi-variant peptide sequences were generated per cell line. Our approach differs from most prior custom database generators, which offer 'Single-Each'[12,52,69,70] or 'All-in-One' outputs[71,72]—for the former, all protein sequences harbor one SAAV each; for the latter, each protein harbors all SAAVs detected. While establishing our combinatorial databases, we observed that a small number of highly polymorphic genes (Supplementary Data 4) markedly increased database size—exemplifying this increased complexity, upwards of 1 billion combinations ($2^n - 1$) are possible for protein sequences with 30 or more SAAVs. To determine the practical limit for the number of SAAVs/protein, we performed test searches where we limited the number of variants to combine (Supplementary Data 4). We find that nearly all variants are retained with databases that include combinations for proteins with up to 25 variants (Supplementary Data 4). For the small set of highly polymorphic protein sequences (e.g., HLA, MUC, and OBSCN (Supplementary Data 4), Single-Each sequences were searched (Fig. 3F).

Next, for all 11 sequenced cell lines (Supplementary Data 3), we prepared and acquired a set of high-coverage cysteine chemoproteomics datasets (Fig. 4A) with the goal of identifying acquired cysteines and variants proximal to reference cysteines. In aggregate, 32,638 total canonical cysteines were identified on 7233 total proteins (Supplementary Fig. 18 and Supplementary Data 4). Two-stage MSFragger search using our sample-specific combinatorial databases identified a total of 59 gained cysteines and 302 SAAVs located proximal to 343 reference cysteines (Fig. 4B and Supplementary Data 4).

Across all these identified SAAVs, we were particularly interested in assessing the impact of our combinatorial exome and RNA-seq SAAV databases on variant identification. We identify six multi-variant-containing peptides (Supplementary Data 4). One noteworthy example is the L86P/F92C peptide from the mitochondrial enzyme HADH, which catalyzes beta-oxidation of fatty acyl-CoAs—two variants, one

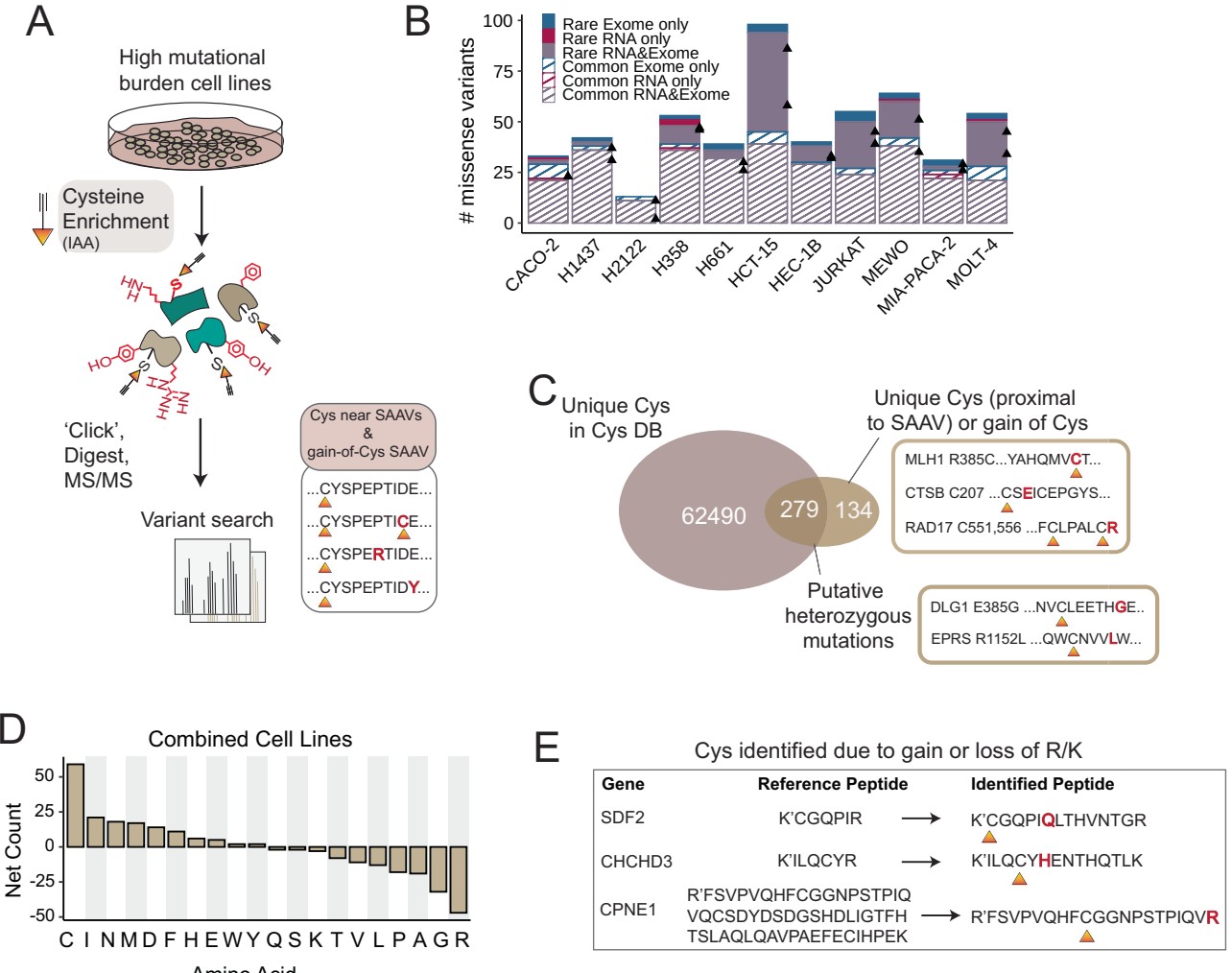

**Fig. 4 | Variant peptide identification on tumor cell lines. A** Cell lysates were labeled with pan-reactive iodoacetamide alkyne (IAA) followed by 'click' conjugation onto biotin azide enrichment handle. Samples were prepared and acquired using our established SP3-FAIMS chemoproteomic platform[31,32,131] single pot solid phase sample preparation (SP3)[132] sample cleanup, neutravidin enrichment, sequence-specific proteolysis, and LC-MS/MS analysis with field asymmetric ion mobility (FAIMS) device[133]. Experimental spectra are searched using a custom FASTA for variant identification. The sample-set includes a reanalysis of previously reported datasets from Yan et al.[31]. (Molt-4, Jurkat, Hec-1B, HCT-15, H661, and H2122 cell line) with newly acquired datasets (H1437, H358, Caco-2, Mia-PaCa-2, and MeWo cell lines). **B** Total numbers of unique missense variants identified from either RNA-seq or WE-seq or both after using two-stage MSFragger search and philosopher validation from duplicate ($n = 2$) datasets; stripe vs solid denotes common and rare variants, black triangles represent replicate total counts, indicated is sequencing source and type of variant. **C** Overlap of identified cysteines from variant searches with cysteines in the CysDB database[29]. **D** Net amino acid changes for all cell lines combined. **E** Example of cysteines identified from loss-of-arginine/lysine peptides. Data is found in Supplementary Data 4, and source data is in the Source Data file.

from RNA-seq and one from exome-seq were detected in this peptide. For the I105V/A114V peptide from enzyme GSTP1, the I105V variants were flagged as bad-quality reads from RNA-seq data but passed filters from the exome-seq data (Supplementary Data 4). Of these combination variants, two are exome-seq-only derived variants that span exon boundaries. While the coverage of these multi-variant peptides is modest, these examples illustrate the value of combinatorial databases for proteogenomic search.

We next investigated the specific features of the identified variants, with the goal of determining if we were capturing cysteines not covered in prior chemoproteomic studies, including those gained due to variant-induced changes to the tryptic peptide landscape. By comparing to our high coverage database of cysteine chemoproteomic data, CysDB[29], we find that chemoproteogenomics identified 74 canonical sequence cysteines located proximal to variants and 60 acquired cysteines that had not been previously reported in CysDB (Fig. 4C). Notable examples of acquired cysteine variants not reported in CysDB

include acquired cysteines KRAS G12C and PRKDC R2899C. Consistent with the aforementioned genomic data findings, we observe arginine as the most frequently lost out of detected Cys-proximal SAAVs (Fig. 4D). We detect 15 total cysteines in peptides that harbor gain/loss-of-arginine that were previously too long or too short to be identified (Fig. 4E and Supplementary Data 4). Exemplifying these peptides, for the cysteine protease cathepsin B (CTSB), we identify Cys207 in HCT-15 cells, which was not identified in CysDB—a K209E mutation that creates a longer tryptic peptide sequence compared to the reference sequence ('CSK' to 'CSEICEPGYSPTYKQDK') (Fig. 3K). Taken together, these examples illustrate how the pronounced loss of arginine can impact the detection of both reference and variant cysteines.

### Chemoproteogenomics identifies both rare and common variants including highly deleterious sites
One of our overarching goals for establishing chemoproteogenomics was to enhance the discovery of likely functional variants. Therefore

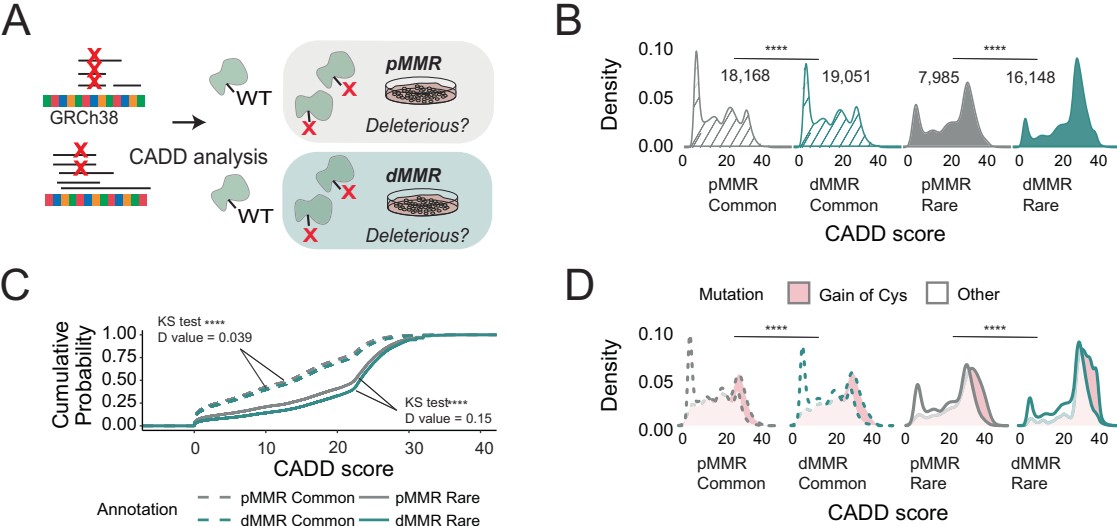

**Fig. 5 | Chemoproteogenomics identifies predicted deleterious sites. A** Scheme of CADD score analysis for two dMMR and non-dMMR cell lines. **B** Distribution of CADD scores for indicated variant grouping; statistical significance was calculated using a two-sided Mann-Whitney U test, ****$p$ <0.0001 (Common, $p$ = 3.1e-16; Rare, $p$ = 5e-46). **C** Empirical cumulative distributions (ECDF) were computed for CADD scores with indicated grouping; statistical significance was calculated using a two-sample Kolmogorov-Smirnov test, ****$p$ <0.0001 (Common, $p$ = 1.7e−12; Rare, $p$ = 6.4e-34). **D** CADD score distributions for gain-of-cysteine separated by grouping; statistical significance between gained cysteine values was calculated using a two-sample Kolmogorov-Smirnov test, ****$p$ <0.0001 (Common, $p$ = 3.1e-6; Rare, $p$ = 6.4e-10). Data is found in Supplementary Data 3, and source data is in the Source Data file.

we next assessed features of chemoproteogenomic-identified SAAVs that could provide insights into the discovery of SAAVs of functional significance. Given the comparatively modest coverage of acquired cysteine SAAVs, relative to the genomic dataset, we opted to analyze both datasets in parallel to delineate specific features that could inform both the likelihood of proteomic detection and residue functionality.

With the goal of parsing features that favor SAAV detection, we next asked whether chemoproteogenomics favored the detection of rare or common variants or those identified in either or both RNA- and exome-sequencing datasets, with the hypothesis that rare variants may be less likely to be expressed at the protein level. We find that the relative proportion of SAAVs identified by chemoproteogenomics (Fig. 4B) largely parallels the trends observed in our sequencing data (Fig. 3B), with higher detection for dMMR cell lines, particularly for rare variants. These trends extend to acquired cysteines, with similar proportions of rare and common cysteine SAAVs identified by both genomic and chemoproteogenomic analysis. Notably, most chemo proteogenomic-detected SAAVs were found in both the exome and RNA-Seq datasets (Fig. 4B), pointing toward the likelihood that variant calling from RNA-seq data should prove sufficient for variant detection.

Towards guiding the discovery of functional SAAVs, we also stratified the predicted deleteriousness of the identified missense variants (Fig. 5A and Supplementary Data 3). We focused on the Combined Annotation- Dependent Depletion (CADD) score due to its highly reported specificity and sensitivity[73] and our prior findings that showed a strong association between cysteine functionality and a high CADD score[35]. Unsurprisingly, our analysis revealed higher CADD scores for rare variants compared to common variants, across the cell line panel (Fig. 5B, C and Supplementary Data 3). More unexpectedly, we observed a marked increase in the predicted pathogenicity of the rare variants detected in dMMR cell lines compared with pMMR cell lines (the top 1% most predicted deleterious mutations have CADD phred-scaled scores >20) (Fig. 5B, C and Supplementary Figs. 19, 20). These trends were maintained in our proteomic datasets, with enrichment of high CADD score missense variants in the dMMR rare

variant subset, including for gain-of-cysteine SAAVs (Supplementary Fig. 21). Even more striking, further stratification by specific gained or lost amino acids (Fig. 5D and Supplementary Figs. 22–25), revealed that gained cysteine missense mutations are the most significantly enriched for high predicted deleterious scores across all pMMR and dMMR cell lines (Supplementary Data 3). These findings provide evidence in support of the use of dMMR cell lines as useful model systems for proteogenomic detection of likely deleterious variants.

Possibly complicating matters, nearly all of the 77 variants in Clinvar and identified by chemoproteogenomics were annotated as benign (Supplementary Data 4). Similarly, chemoproteogenomics failed to capture several key Census gene SAAVs that we detected on the genomic level (e.g., *SMAD4* (D351H) in CaCo-2, *FBXWY* (R505C) in Jurkat and *CDK6* (R220C) in Molt-4 cells). These examples provide additional anecdotal evidence of the challenges associated with detecting deleterious variants.

Chemoproteogenomics did, however, capture 16 mutations and 7 putative driver mutations (dN/dS p-values) that were identified in Census genes. Several high-value census gene SAAVs were distinguished by both high CADD scores (>20) and proximity to known pathogenic mutation sites. These variants of interest include MLH1 R385C, RAD17 L557R (proximal Cys551/556), MSN R180C, HIF1A S790N (proximal Cys800) and CTCF R320C, a likely pathogenic position in this protein (Supplementary Data 4). A prevalent driver was KRAS G12C, which was identified in several of the cell lines known to harbor this variant as a driver mutation (MIA-PACA-2 and H358 but not H2122). As KRAS expression is known to vary across cell lines[58], this data suggests both H358 and MIA-PACA-2 cell lines are suitable for chemoproteogenomic target engagement analysis of G12C-directed compounds.

## Chemoproteogenomics captures previously undetected variants

Exemplifying the utility of chemoproteogenomics (Fig. 6A) to uncover previously undetected variants, we find that 20 of the identified SAAVs have not been previously reported in COSMIC, CCLE, or ClinVar (Supplementary Data 4). One variant of unknown significance, not

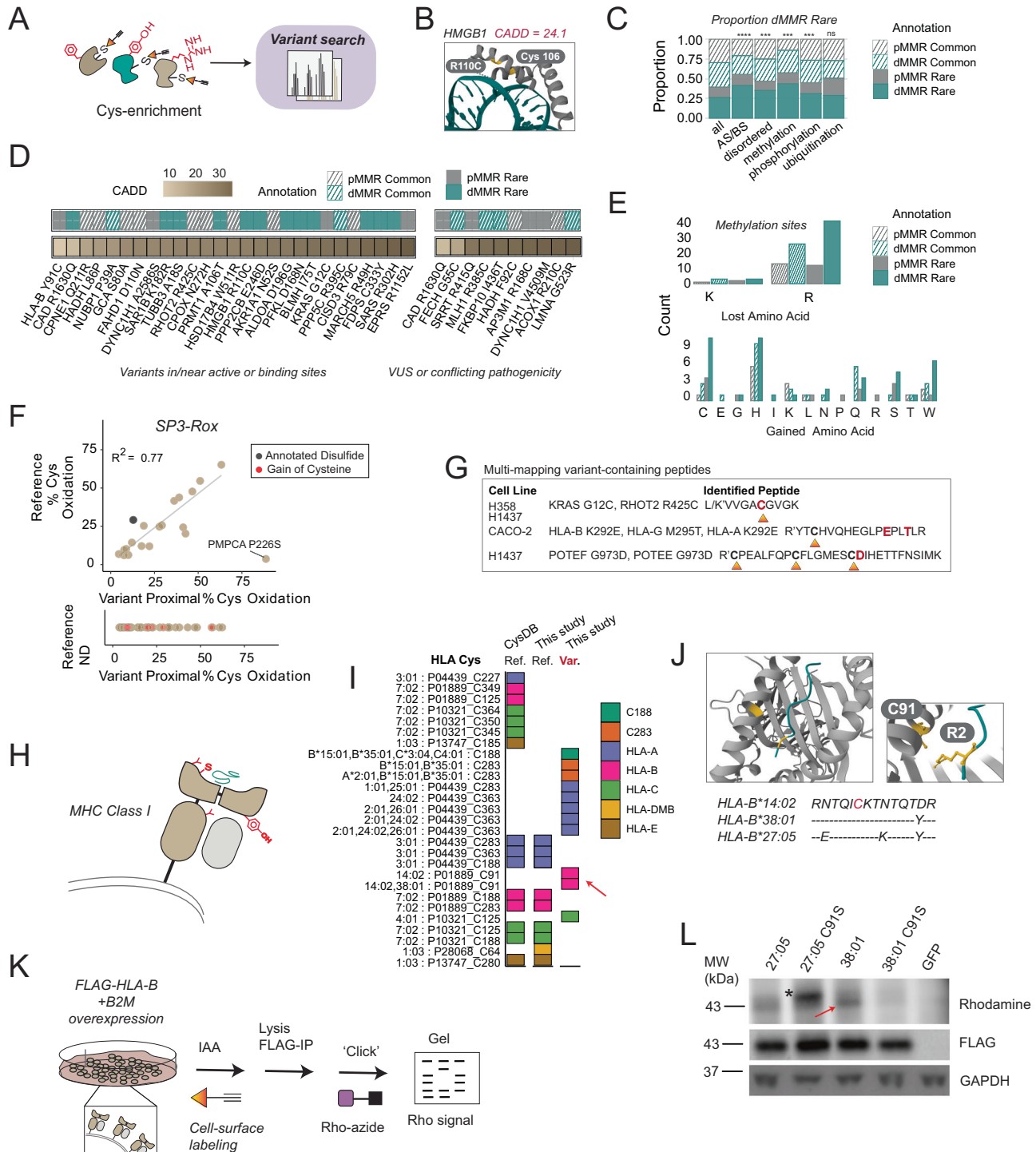

**Fig. 6 | Chemoproteogenomics identifies SAAVs proximal to likely functional sites. A** Scheme for chemoproteomics data search to identify variants from duplicates (*n* = 2). **B** Crystal structure of HMGB1 indicating detected Cys110 and nearby Cys106 (*yellow*) (PDB: 6CIL). **C** Proportion of variants belonging to the indicated sites; AS/BS = in or near active site/binding site in genomics data as annotated by UniProtKB or Phosphosite; statistical significance calculated using the two-sample test of proportions, *** *p* <0.001, **** *p* <0.0001, ns *p* >0.05.**D**) Chemoproteogenomic-identified variants identified in or near active and binding sites with CADD score, common/rare, cell line dMMR/pMMR annotations. **E** Amino acid changes at protein methylation sites as identified by Phosphosite from genomics data. **F** Re-analysis of SP3-Rox[24] oxidation state data in Jurkat cells (*n* = 6) acquired cysteines and 54 variants proximal to acquired cysteines. **G** Example of cysteines identified from loss-of-arginine/lysine peptides. **H** Schematic of highly variable HLA binding pocket containing cysteine with bound peptide. **I** Coverage of HLA cysteines from this study and in CysDB; color indicates HLA type or multi-mapped cysteines. **J** Crystal structure of HLA-B 14:02 (PDB: 3BXN) with highlighted Cys67 and Arg P2 position of bound peptide; alignments of Cys91 regions of three HLA-B alleles. **K** Workflow to visualize HLA cysteine labeling; first cells were harvested and treated with IAA followed by lysis, FLAG immunoprecipitation, and click onto rhodamine-azide. **L** Cys-dependent cell surface labeling of HLA-B alleles with IAA, the band indicated with a red arrow and non-specific band represented with an asterisk (representative of 2 two biological replicates). Data is found in Supplementary Data 3 and Supplementary Data 4, and source data is in the Source Data file.

reported in ClinVar, is high mobility group box 1 (HMGB1) R110C labeled in the Molt-4 cell line (Fig. 6B) (CADD score = 24.1). Adjacent Cys106 is a cysteine under a highly controlled redox state that is responsible for inactivating the immunostimulatory state of HMGB1[74–77]. We also identify Cullin-associated NEDD8-dissociated protein 1 (CAND1) G1069C—a site which mutated in the *Arabidopsis thaliana* ortholog reduces auxin response[78]—and SARS R302H (proximal Cys300;CADD = 32), a mutation in the ATP binding site of serine-tRNA ligase, which is a tRNA ligase involved in negative regulation of VEGFA expression[79]. These three examples illustrate the capacity of chemoproteogenomics for the identification of potentially functionally relevant variants.

## Chemoproteogenomics identifies SAAVs proximal to likely functional sites

As CADD scores only provide a prediction of deleteriousness, we also asked whether any of the identified variants are located proximal to known functional sites and sites of post-translational modification. At the genomic level, we find that the dMMR rare variant set is enriched for known proximal active site/binding site residues (Fig. 6C and Supplementary Data 3). Within the proteomic dataset, only 3 variants were located at annotated active or binding sites including previously mentioned HMGB1 R110C, tRNA synthetase EPRS R1152L (proximal Cys1148; CADD = 33), a mutation known to cause complete loss of tRNA glutamate-proline ligase activity[80], and SARS R302H. Thus, we broadened our analysis to include SAAVs at or proximal to UniProtKB annotated active sites (AS) and binding sites (BS) (Fig. 6D). We find that 27 SAAVs are located within the permissive range of 10 amino acids of a known functional residue, including 4 active sites and 24 binding sites.

Beyond AS/BS proximity, we also assessed proximity to other likely functional sites, known functional domains, and PTM-modified sites reported by Phosphosite[81]. We find generally no marked bias for variants located in specific domain types, with the ubiquitous P-loop NTPase domain as the most SAAV-rich domain (Supplementary Fig. 27 and Supplementary Data 4). We do, however, observe that variants in GPCR transmembrane domains are likely challenging to detect by proteogenomics. In our genomic datasets, GPCR transmembrane domains are enriched for variants. This enrichment does not extend to our proteomic analysis (Supplementary Fig. 27 and Supplementary Data 4). This difference in coverage can be rationalized by membrane proteins' generally low abundance, hydrophobicity, and the lack of tryptic sites in transmembrane domains, which together make proteomic detection of peptides from GPCRs and related proteins particularly challenging[9,82,83].

Intriguingly, analysis of known PTM-modified sites reported by Phosphosite[81] revealed a significant association between arginine methylation sites and rare variants in dMMR cell lines (Fig. 6E). Examples of such variants that we detected via chemoproteogenomics include the methylation sites XRN2_p.R925C (CADD = 31) and HSPH1_p.R265C (CADD = 32), as well as phosphorylation site CNN2_p.S244Y (CADD = 27.5). These findings are consistent with loss-of-arginine as a frequent consequence of exonic CpG mutability[64,84], together with the roles of MMR in protecting against CpG-associated deamination[85]. As 60% of the gained cysteines in our data resulted from loss-of-arginine (Supplementary Fig. 26), we expected that many of these variants would result in altered PTM status.

Because cysteines play critical roles in protein structure via disulfide bond formation together with additional cysteine oxidative modifications[86], we asked whether identified loss-of-cysteine variants (10 in total) were annotated as involved in disulfides. Likely due to the comparatively small number of loss-of-cysteine variants, none were observed with disulfide annotations. To further pinpoint whether any variants are sensitive to oxidative modification, we subjected our previously reported Jurkat cell redox chemoproteomics datasets to reanalysis[24]. For nearly all of the cysteines quantified with proximal

variants, both in our reference database searches and second stage searches, we observed a high concordance between variant- and reference sequence oxidation ($R^2 = 0.77$). One notable exception was the Mitochondrial-processing peptidase enzyme (PMPCA) Cys225, where markedly different cysteine oxidation states were measured for the reference peptide cysteine (~3% oxidation) and variant peptide cysteine (~88% oxidation) (Fig. 6F). These data provide evidence that the proximal P226S (CADD = 25.1) mutation profoundly impacts Cys225 sensitivity to oxidative modifiers.

## Chemoproteogenomics enables the high confidence detection of multi-mapped genes, including for highly polymorphic sequences

One challenge for chemoproteogenomics is the accurate assignment of variant-containing peptide sequences to the corresponding mutated gene. Exemplifying this challenge, and as a cautionary example in mapping peptides, we identify several SAAV-peptides that match to multiple protein sequences, including sequences in human leukocyte antigens (HLA) and POTE ankyrin domain family proteins (Fig. 6G). Most notably, the RHOT2 R425C (CADD = 23.2) mitochondrial GTPase peptides in H358 cells have exact sequence similarity to KRAS G12C peptides; these half-tryptic peptides are also identified in H1437 cells that do not harbor the KRAS G12C variant. Thus, without cell-line matched variant databases, chemoproteomic data for the RHOT2 cysteine could easily be misconstrued as reflective of the G12C KRAS peptide.

The HLA or Major Histocompatibility Complex (MHC) Class I molecules represent another particularly challenging class of sequences for chemoproteogenomic analysis, distinguished by the presence of multiple possible variant combinations and high sequence redundancy. HLA are highly polymorphic, with ~15,000 HLA alleles reported in the human population[87]. Exemplifying the impact of this polymorphism on proteomic sequence coverage, our panel of cell lines alone harbor >25 HLA-A, B, and C alleles (Supplementary Data 3), while most protein reference databases only contain one copy of each MHC Class I and Class II molecule. This complexity together with the important functions in innate immunity and therapeutic relevance of the HLA proteins[88–91] inspired us to assess the impact of chemoproteogenomics on achieving improved coverage of highly polymorphic genes (Fig. 6H).

Demonstrating the value of our proteogenomic analysis, we achieved ~50% more coverage of HLA-A sequence in comparison to reference searches (Fig. 6I, Supplementary Fig. 28). A key finding of our analysis was detection of HLA-B Y91C (CADD = 4.9) (C67 post signal peptide cleavage), which lies in the extracellular peptide binding pocket of HLA-B and was identified as IAA-labeled in MeWo cells (Fig. 6J). The MeWo cell line HLA alleles (HLA-B*14:02 and HLA-B*38:01) both harbor this comparatively rare Cys. Notably, this cysteine is also a key feature of the pathogenic ankylosing spondylitis associated allele HLA-B*27(Brewerton et al. 1973; Alvarez et al. 2001).

To further vet the capacity of our chemoproteogenomic platform in faithfully capturing cysteine peptides from multi-mapped genes, we established a gel-based activity-based protein profiling (ABPP)[92–94] platform for Cys67 HLA alleles. We co-expressed C-terminal FLAG tagged HLA-B*38:01 and the related and pathogenic HLA-B*27:05 alleles with beta-2-microglobulin (β2m) and subjected cells to in situ IAA labeling followed by lysis, FLAG immunoprecipitation to enhance the detectability of the HLA cysteine and click conjugation to rhodamine azide (Fig. 6K). Gratifyingly, we observed a Cys67-specific rhodamine signal (Fig. 6L) that was blocked by the Cys67Ser point mutation, showcasing the utility of gel-based ABPP in visualizing HLA small molecule interactions. Notably IAA labeling was also observed for HLA-B27:05, although the presence of a strong co-migrating band in the HLA-B27:05 C67S immunoprecipitated sample complicates interpretation of the specificity of this labeling to Cys67. We were

unable to observe comparable signal in lysate-based labeling studies, supporting enhanced accessibility of this cysteine to cell-based labeling (Supplementary Fig. 29).

## Assessing how differential expression impacts chemoproteogenomic detection

Our comparatively modest coverage of SAAVs achieved by chemoproteogenomics (particularly when compared to our genomics datasets) is on par with the coverage reported by most prior proteogenomics studies[6,8,17]. A notable exception is a recent study by Coon and colleagues that implemented ultra-deep fractionation to achieve more global coverage of variants[9]. Inspired by this work, we next sought to ask whether chemoproteogenomics, with its built-in enrichment step, would enable sampling of variants not detectable by fractionation methods (Fig. 7A). We subjected lysates from HCT-15 and Molt-4 cells, which were chosen based on high rare missense burden, to tryptic digestion, off-line high pH fractionation and LC-MS/MS analysis. In aggregate across both cell lines, we identified 8,435 proteins and 149,006 peptides, including 1069 unique SAAVs found in 1352 total peptides using our two-stage FDR MSFragger search (Fig. 7B, Supplementary Fig. 30 and Supplementary Data 5). Illustrating the use of our combinatorial databases, 26 peptides were identified that contained multiple variants (Fig. 3F and Supplementary Data 5).

With these bulk datasets in hand, we next compared the variant content to that afforded by chemoproteogenomics for the matched HCT-15 and Molt-4 proteomes (145 total SAAVs identified by chemoproteogenomics for these two cell lines). Net gained amino acid analysis (Supplementary Figs. 31, 32) revealed similar trends, with cysteine in the top three gained and arginine as the most lost amino acid for both enriched and unenriched datasets. Illustrating the added value of chemoproteogenomics, 70 SAAVs, including eight acquired cysteines, were uniquely identified compared to unenriched datasets (Fig. 7C and Supplementary Data 4, 5). Furthermore, we find that enrichment afforded a ~5-fold boost in the relative fraction of acquired cysteines captured (Fig. 7D). Alongside the benefits of chemoproteomics capture, bulk proteomic analysis revealed unique variants. Bulk analysis identified 85 notable variants belonging to Census genes, including BRD4 E451G (CADD = 31) and KRAS G13D (CADD = 23.8), and 26 rare/common variants of uncertain significance in ClinVar, including rare gain-of-cysteines ubiquitin hydrolase USP8 Y1040C (CADD = 28.5) and LMNA R298C (CADD = 27.2) (Fig. 7E and Supplementary Data 5). Most of these census variants are found in peptides not containing cysteines and thus, should not be detected by chemoproteogenomics.

Given that cysteine chemoproteomics requires peptide derivatization with a comparatively large (463 Da) biotin modification, we additionally postulated that some differences in coverage could also be ascribed to the behavior of peptides during sample acquisition. Comparing the properties of the SAAV peptides detected by chemoproteogenomics versus proteogenomics, we observed a more restricted charge state distribution for cysteine-enriched samples and no appreciable differences in the amino acid content beyond enrichment for cysteine (Supplementary Fig. 33). While we did not observe differences in the peptide lengths in our comparison between the chemoproteomic-enriched and high pH detected SAAV peptides, a marked significant increase in SAAV peptide length (average 5AA longer) was observed compared to reference peptides in both datasets (Fig. 7F). This increased peptide length is consistent with the ubiquity of loss-of-arginine SAAVs in both datasets, which are favored in the longer length peptides (Supplementary Fig. 34). Thus, we concluded that chemical properties are not the primary reason for the difference in coverage between bulk and cysteine enrichment proteomics.

Therefore, we asked whether protein or RNA abundance might rationalize the differences in SAAV coverage for each method. Comparison of normalized transcript counts for SAAV-matched genes identified either by chemoproteogenomics or in our bulk proteomic dataset, for HCT-15 cells revealed no significant difference between measured transcript abundance between the sets (Fig. 7G and Supplementary Data 5). A Supplementary Data subset of SAAVs (3262 total, including PIK3CA E545K, TP53 S241F, SMARCA4 R885C TCGA hotspot mutations all with CADD >27) with low abundance transcripts (less than 4000 normalized counts) were not detected in either the chemoproteogenomics or bulk proteogenomics. This finding provides evidence that low transcript abundance correlates with a decreased likelihood of variant detection both for bulk proteomics and for chemoproteomics. These trends for relative ease of proteomic detection are not restricted to variants and also extend to reference cysteines, with a marked enrichment of undetected cysteines encoded by low abundance transcripts (Supplementary Fig. 35).

Given the likely disconnect between transcript abundance and protein abundance[95–97] for some SAAVs analyzed, we also extended these analyses to measures of protein abundance. Using label-free quantification (LFQ) analysis, no difference was observed in protein abundance, inferred from quantified protein intensities, between the bulk fractionated samples and the chemoproteogenomic samples (Fig. 7H and Supplementary Data 5). Consistent with low abundance protein variants being challenging to detect, SAAVs detected via both proteomics workflows were observed to belong to more abundant proteins, in comparison to variants only detected via genomics.

Both the transcript and protein abundance analyses do not delineate reference from variant-specific transcript/protein sequences. Therefore, to further delineate the capacity of chemoproteogenomics to detect low abundance variants, we assessed the variant allele frequencies (VAF) for detected SAAVs. We find that high-pH variant allele frequencies (VAF) were significantly higher than the chemoproteogenomic detected SAAVs, including the acquired cysteine subset, which were comparable to the aggregate bulk RNA-seq VAFs (Fig. 7I, Supplementary Data 5 and Supplementary Fig. 34). This enrichment for lower VAF for the chemoproteogenomic detected SAAVs hints at the utility of chemoproteogenomics for capture of rare variant-containing peptides.

Guided by these findings, we asked whether chemoproteogenomics was well suited to capture deleterious variants, with the hypothesis that proteins harboring these likely damaging variants may be lowly expressed. Consistent with this premise, the mean CADD scores for the chemoproteogenomics identified variants were significantly higher than those calculated for the variants identified via bulk proteomics (Supplementary Fig. 36). Notable high-CADD score (>29) variants identified only from enrichment include lysine demethylase KDM3B D1444Y, RNA polymerase POLRMT R805C, glycoprotein transporter LMAN2 R218C and Serine/threonine-protein phosphatase PP1-alpha catalytic subunit PPP1CA D203N (Fig. 7C). Taken together these findings illustrate the added value of chemoproteogenomics in capturing functionally interesting variants.

## Chemoproteogenomics enables ligandability screening

As demonstrated by our previous studies, cysteine chemoproteomics platforms are capable of pinpointing small-molecule targetable cysteine residues[21,30,31,34]. Therefore, we next paired our two-stage FDR search method with cysteine-reactive small molecule ligandability analysis to establish a chemoproteogenomic small molecule screening platform (Fig. 8A). We first opted to use the widely employed scout fragment KB02[21] (Fig. 8B) to compare the ligandable variant proteomes for three high variant burden dMMR cell lines (HCT-15, Jurkat, and Molt-4). For KB02 treated samples, we identified 210 total variants, of which 8 were ligandable (Fig. 8C). The high concordance for ratios detected for variant peptides with multiple alleles provides evidence of the robustness of our platform and hints that most cysteine proximal variants do not substantially alter cysteine ligandability (Fig. 8D).

To provide a focused assessment of the structure-activity relationship (SAR) of small molecules for individual cysteines, we next

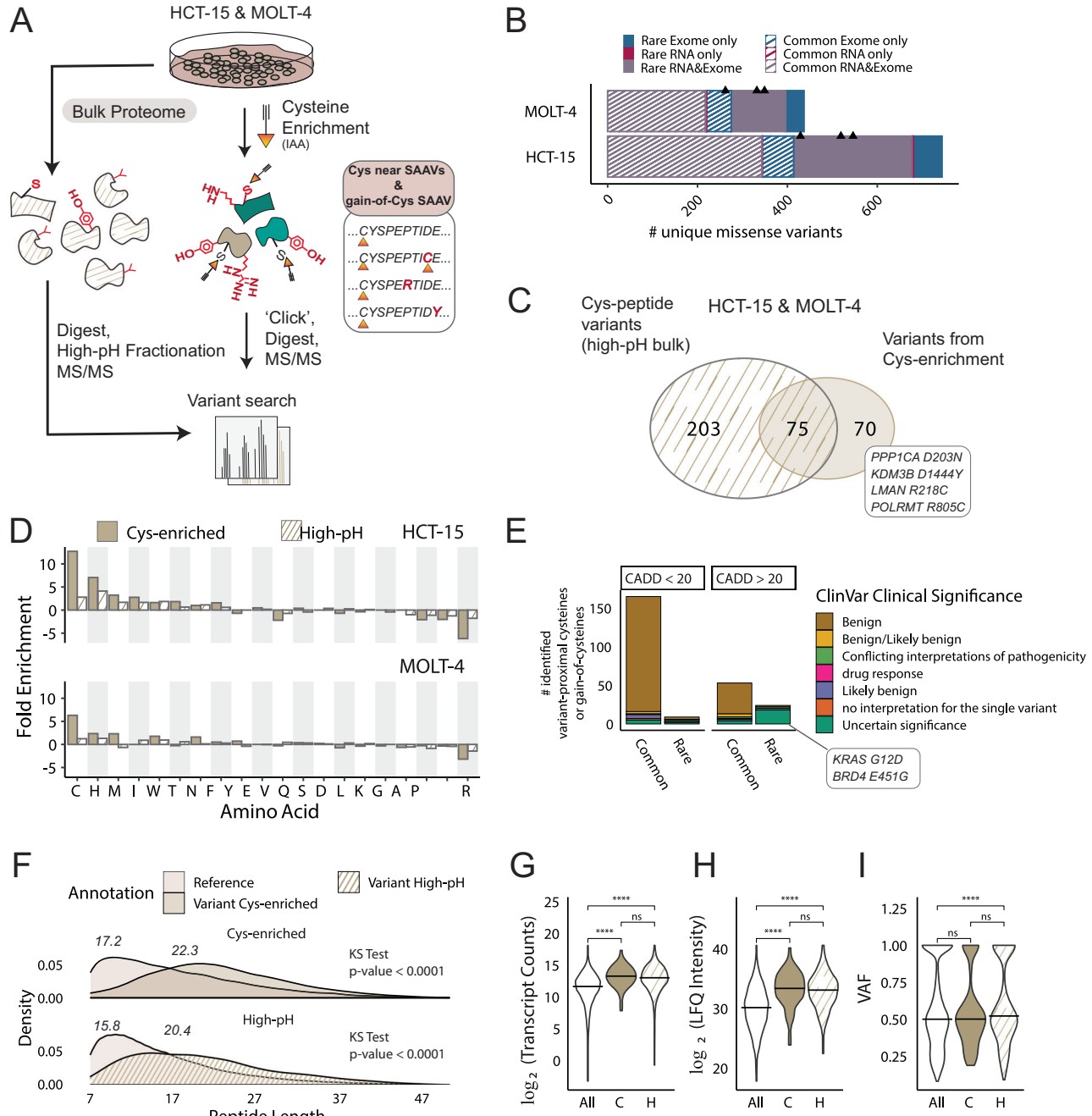

**Fig. 7 | Comparison of variants identified from cysteine enrichment and bulk proteomics. A** Workflow for high-pH fractionation of lysates. Cell lysates are treated with DTT and iodoacetamide followed by digestion, high-pH fractionation, and LC-MS/MS analysis. Triplicate high-pH sets ($n = 3$) for HCT-15 and Molt-4 cells were used. **B** Total numbers of unique missense variants identified from either RNA-seq or WE-seq or both after using a two-stage MSFragger search of high-pH datasets, black triangles represent replicate total counts. **C** Overlap of cysteine-containing peptide variants identified from bulk fractionation and cysteine enrichment datasets. **D** Fold enrichment of amino acids as a ratio of the net amino acid frequency (gain minus loss) to the amino acid frequency in all missense-containing proteins detected in high-pH and cys-enriched datasets. **E** High-pH detected variants stratified by CADD score and ClinVar clinical significance. **F** Peptide lengths of reference and variant peptides identified in dataset types,

statistical significance using two-sample Kolmogorov-Smirnov tests, ****$p$ <0.0001. **G** DE-seq normalized transcript counts for all RNA variants. 'All', variants detected from cys-enrichment 'C', and variants detected from high-pH fractionation 'H' in HCT-15 cells; bar indicates the mean value (All vs C, $p = 7e$-17; C vs H, $p = 0.17$; All vs H, $p <2e$-16). **H** Label-free quantitation (LFQ) intensities for proteins matched to all RNA variants 'All', variants detected from cys-enrichment 'C', and variants detected from high-pH fractionation 'H' in HCT-15 cells; bar indicates the mean value (All vs C, $p <2e$-16; C vs H, $p = 0.19$; All vs H, $p <2e$-16). **I** Variant allele frequencies (VAF) (total reads/total coverage per site) for RNA-seq variants called in HCT-15 and Molt-4 cells (All vs C,$p = 0.74$; C vs H, $p = 0.053$; All vs H, $p = 9e$-5). **G–I** bar indicates the median, statistical significance was calculated using two-sample Kolmogorov-Smirnov tests, ****$p$ <0.0001, ns $p$ >0.05. Data is found in Supplementary Data 5, and source data is in the Source Data file.

subjected the HCT-15 proteome to more in-depth analysis using a small panel of custom electrophilic fragments (Fig. 8B and Supplementary Fig. 37). We observed 27 total liganded variant peptides in 27 proteins in the HCT-15 proteome, which were labeled by one or more

compounds (Fig. 8C). As with the KB02 cell line comparison, nearly all multi-allelic peptides showed comparable ratios (Fig. 8E). One notable exception was EPRS P1482T (CADD = 27.2), which showed markedly different reference and variant ratios—the mutated proline nearby Cys

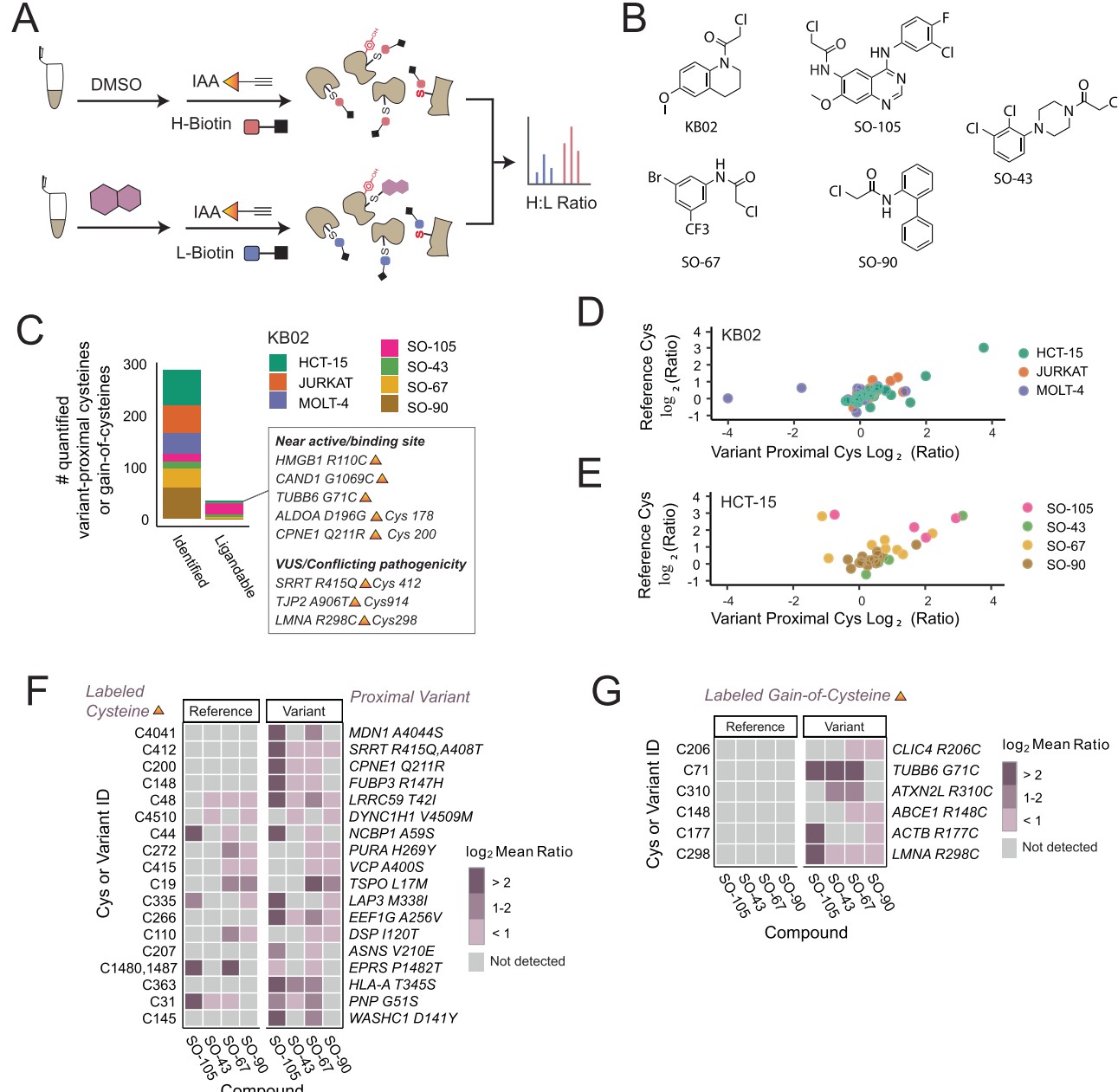

**Fig. 8 | Assessing ligandability of variant proximal cysteines and gain-of-cysteines. A** Schematic of activity-based screening of cysteine reactive compounds; cell lysates are labeled with compound or DMSO followed by chase with IAA and 'click' conjugation to heavy or light biotin click conjugation to our iso-topically differentiated heavy and light biotin-azide reagents, tryptic digest, LC-MS/MS acquisition, and MSFragger analysis. **B** Chloroacetamide compound library. **C** Total quantified variants and total ligandable variants (log₂ Ratio >2) identified stratified by cell line (KB02 data) or compound (HCT-15 cell line). **D** Correlation of high-confidence variant containing and reference cysteine ratio values from KB02 data. **E** Correlation of high-confidence variant containing and reference cysteine ratio values from SO compound data. **F** Log₂ heavy to light ratio values for variant containing and reference cysteine peptides. **G** Subset of gain-of-cysteine peptide variant log₂ ratios. Data is found in Supplementary Data 6, and source data is in the Source Data file.

1480 may be requisite for labeling by electrophilic fragments (Fig. 8F). As multi-allelic acquired cysteine sites cannot be captured sans cysteine, no analogous ratio comparisons could be performed for the 6 total quantified acquired cysteines (Fig. 8G).

We also asked whether any of the ligandable variants would likely alter protein activity. We chose to focus on three metrics to guide our prioritization of likely variants for functional analysis, CADD score, proximity to known functional sites, and variants that result in gained cysteines. We analyzed active site and binding sites within 10 angstrom distance of the ligandable cysteine residues and cysteine-proximal variant sites (Supplementary Data 6). We find three ligandable cysteines near or in active/binding sites including previously identified

HMGB1 Cys106 (R110C, CADD = 24.1) (Fig. 8A), as well as Aldolase A ALDOA Cys178 (G196G, CADD = 26.2) and HLA-B/C Cys125 (V127L/ S123Y, CADD <1). Other notable sites were the aforementioned CAND1 G1069C and Tubulin beta 6 (TUBB6) G71C, CADD = 32, which resides proximal to the GTP binding site (Fig. 8C, G).

Of these intriguing variants, we selected CAND1 and HMGB1 for follow-up analysis. For each protein, we generated both the corresponding gain-of-cysteine mutations together with tryptophan mutations. Our prior work[98] and that of others[99] have shown the comparatively bulky tryptophan mutation serves as a useful surrogate for small molecule binding. Therefore, as our scout fragments are modestly potent, we chose to use tryptophan point mutations in lieu of

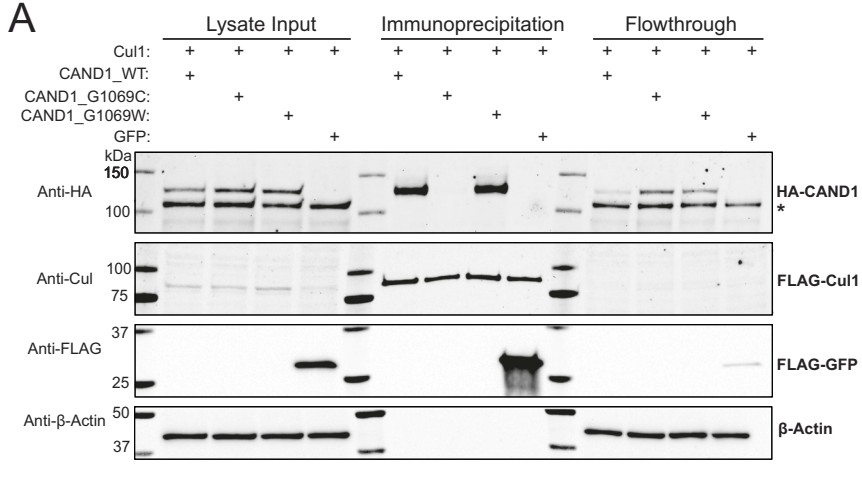

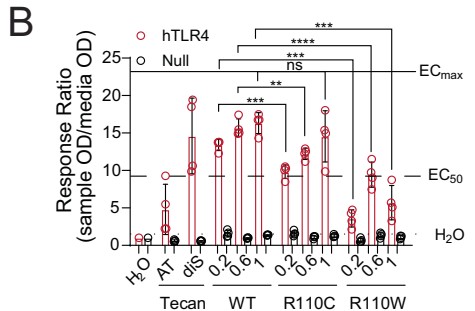

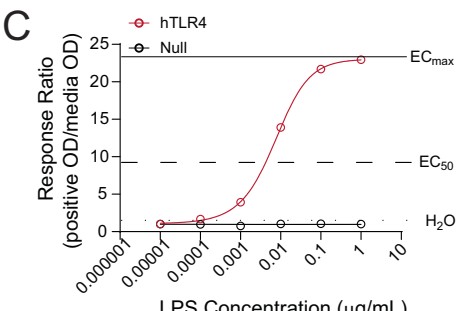

**Fig. 9 | Functional studies of CAND1 and HMGB1. A** WT and G1069C mutant CAND1 proteins bind Cul1 while the G1069C CAND1 mutation perturbs binding. HEK293T cells were co-transfected with FLAG-Cul1 and the given HA-tagged CAND1 protein (WT, G1069C, or G1069W) or control FLAG-GFP. Anti-FLAG resin was used to pull down FLAG-Cul1 from cell lysates along with any complexed proteins. Western Blots were incubated with the indicated primary antibodies, *indicates a non-specific HA band. **B** HMGB1 proteins were tested for the ability to induce TLR4-mediated immune response using HEK-Blue reporter cell lines (hTLR4 and Null control) and corresponding PRR assay. Results show mean response ratios (error bars = SD, $n = 4$ per condition) of hTLR4 and Null cells to increasing concentrations (µg/mL) of WT, R110C, and R110W proteins as indicated over 2 independent experiments. AT = commercially available all-thiol fully

reduced HMGB1; diS = commercially available disulfide HMGB1; working concentration of 0.2 µg/mL for both. Reference lines on the graph indicate $EC_{max}$ (solid line), $EC_{50}$ (dashed line), and $H_2O$ control (dotted line) response ratios to canonical positive control ligand (LPS) specific to the hTLR4 cell line. Significance determined via unpaired two-tailed student's $t$ test; ** = $p < 0.01$, *** = $p < 0.001$, **** = $p < 0.0001$. TLR4 200 ng/mL (WT vs R110C, $p = 0.009$; WT vs R110W, $p < 0.0001$); TLR4 600 ng/mL (WT vs R110C, $p = 0.009$; WT vs R110W, $p < 0.0001$). **C** Response ratio curve of hTLR4 and Null cells to positive control ligand (LPS). EC values are generated using nonlinear regression (Asymmetric (five parameters), X is concentration). For (**A**), western blot data are representative of three independent measurements. Data is found in Supplementary Data 6, and source data is in the Source Data file.

small molecule treatment to minimize the risk of non-specific compound labeling complicating the interpretation of variant functionality. Using a coimmunoprecipitation assay, we find that CAND1 G1069C but not G1069W completely blocks interactions with CUL1 (Fig. 9A). This finding is notable given the important functions of this hairpin in mediating SKP1-SKP2 dissociation from SCF, which is critical to regulating the functions and composition of E3 ligase complexes[100,101].

As a second case study, we turned to HMGB1, which is known to function as a redox-active cytokine[74–76]. Therefore, we opted to assess its binding to toll-like receptor (TLR) 4, which has previously been reported as bound specifically by the disulfide (Cys23-Cys45) form of HMGB1—the fully reduced (all thiol) protein does not activate TLR4 signaling activity. Notably the fully oxidized (including Cys106) form of HMGB1 is also inactive[77]. Thus, we hypothesized that the R110C mutation we identified would decrease cytokine activity. To test this hypothesis, we expressed and purified recombinant wild-type HMGB1 together with both the R110C and R110W mutant proteins. Then using a human TLR4 HEK-Blue reporter cell line[74–76], we compared the relative TLR4 response to treatment with each protein. Providing evidence that our HMGB1 protein is active in this assay, we observe no significant

difference relative to commercially available (TECAN) disulfide (diS) protein and our wild-type protein (Fig. 9B, C). Revealing the functional impact of the R110 mutations, we find that both the acquired cysteine and bulkier tryptophan scanning mutation significantly attenuate HMGB1-induced TLR4 response, with a more substantial effect observed for the tryptophan mutation. Taken together, these two case studies illustrate the utility of chemoproteogenomics in the discovery of functionally important gain-of-cysteine variants.

## Discussion

SAAvs are a ubiquitous feature of human proteins, which remain under-sampled in established proteomics pipelines. Guided by the unique chemistry of the cysteine thiol, we focused our studies on guiding the discovery of functional, redox-sensitive, and potentially druggable acquired cysteine SAAvs together with variants proximal to reference cysteines. To enable the discovery of the variant cysteinome, we merged genomics with mass spectrometry-based chemoproteomics to establish chemoproteogenomics as an integrated platform tailored to capture and functionally assess the missense variant cysteinome. Our chemoproteogenomics study is distinguished by a number of features including: (1) genomic stratification of the

predicted pathogenicity of acquired cysteine residues, (2) cell-line paired custom combinatorial search databases, (3) FragPipe enabled two-stage FDR database search platform ensuring class-specific FDR estimation, and (4) capacity to pinpoint both redox-sensitive and ligandable genetic variants proteome-wide. To facilitate widespread adoption of our approach, including for applications beyond the study of the variant cysteinome, the user-friendly GUI-based FragPipe platform now features a robust semi-automated version of our two-stage FDR search (Supplementary Fig. 2). In total, across 11 cell lines sequenced, chemoproteogenomics identified 1453 missense variants, of which 116 led to gain-of-cysteine.

Our paired genomic and proteomic analyses revealed a number of noteworthy findings that we expect should help to guide ongoing and future efforts in pinpointing functional and therapeutically relevant variants. We find that cysteine acquisition is a ubiquitous feature of human genetic variation spanning rare and common variants identified in COSMIC, ClinVar, and dbSNP. The instability of CpG motifs is a key driver of bulk cysteine acquisition, which occurs largely hand-in-hand with bulk arginine depletion, across both cancer genomes and healthy genomes We also find that the previously reported widespread cysteine acquisition in cancer genomes[38,102] is predominated by mismatch repair deficient cell lines, and particularly MSI high colorectal cancer cell lines. Nearly all of the acquired residues in these lines are not driver mutations, which complicates their use as models for assessing the potential druggability of variants with established clinical connections. Further showcasing the challenges with identifying acquired cysteine driver mutations, we find that many such variants were only detectable at the genomic and not proteomic levels, for the cell lines analyzed, including for both bulk proteomic and chemoproteomic analyses. Thus, we expect that future studies focused on these high-value variants will benefit from targeted approaches, such as GoDIG[103], and together with CRISPR-Cas9 base editing to engineer variants of interest into endogenous loci[28,104–107]. Furthermore, we expect that a subset of the tough-to-detect variants may cause decreased protein stability and premature degradation. Such variant-induced changes in protein half-life may be detectable by pairing proteasome inhibition with chemoproteogenomic analysis.

Our work highlights the potential synergy between chemoproteogenomics, small molecule screening, and redox biology. Our discovery of a cysteine in PMPCA that exhibits variant-dependent changes in oxidation provides an intriguing anecdotal example that supports the future utility of chemoproteogenomics in more broadly characterizing the missense variant redox proteome. Given the critical role that disulfides play in protein structure and folding and the causal roles for cysteine mutations in human disease, for example, the NOTCH mutations that cause the neurodegenerative disorder CADASIL[108], we expect a subset of these lost cysteines could be implicated in altered protein abundance or activity. Through cysteine chemoproteomic capture, we identified ligandable variant-proximal cysteines in Census genes such as RAD17, including one gain-of-cysteine of uncertain significance in LMNA (R298C). Other liganded cysteines proximal to variants of uncertain significance include TJP2 (A906R) and SRRT (R415Q). Demonstrating the utility of our approach, we identified a Cys91 (Cys67) as labeled by IAA both by proteomics and gel-based ABPP. As this cysteine is shared with the pathogenic HLA-B27, it is exciting to speculate about the impact of covalent modification on HLA peptide presentation. Our application of chemoproteogenomics to screening of a focused library of electrophilic compounds identified 32 ligandable variant-proximal cysteines which demonstrates that cysteine ligandability can be assessed proteome-wide in a proteoform-specific manner. Consistent with the significance of the identified variants our functional studies revealed that both HMGB1 and CAND1 variants substantially impact protein activity.

In planning for future enhancement of chemoproteogenomics, we expect that the use of tumor-normal paired variant calling with tools such as MuTect2[109] will further decrease the likelihood of false discovery introduced by factors such as cell heterogeneity and low read quality−for cell lines that lack matched normal controls, we expect that the pairing of publically available datasets (e.g., DepMap, https://depmap.org/) with custom sequencing data, will prove another useful strategy to further bolster the quality and accessibility of variant-containing databases. Such multi-pronged approaches will likely prove most useful when paired with combinatorial custom databases, such as the peptide-based databases reported here, which were designed to minimize increased search space complexity while also more fully accounting for cell heterogeneity.

Looking beyond our current study, we anticipate multiple high-value applications for chemoproteogenomics. Application to immunopeptidomics should uncover additional covalent neoantigen sites, analogous to the recent reports for Gly12Cys KRAS[90,110]. The pairing of chemoproteogenomics with ultra-deep offline fractionation should further increase coverage and allow delineation of variants that alter protein stability, including the numerous high CADD score acquired cysteines, which we find were underrepresented in our proteomics analysis when compared to genomic identification. Inclusion of genetic variants beyond SAAVs will allow for the capture of additional therapeutically relevant targets that result from indels, alternative splicing[4,15,111], translocations, transversions, or even undiscovered open reading frames such as microproteins[112,113]. Thus, chemoproteogenomics is poised to guide the discovery of proteoform-directed therapeutics.

## Methods

Experimental details and Supplementary Data 1–7 can be found in the Supporting Information.

### Cell culture and preparation of cell lysates

Cell culture reagents including Dulbecco's phosphate-buffered saline (DPBS), Dulbecco's modified Eagle's medium (DMEM)/high glucose media, Eagle's Minimum Essential Medium (EMEM), Roswell Park Memorial Institute (RPMI) media, trypsin-EDTA, and penicillin/streptomycin (Pen/Strep), and Horse Serum, heat-inactivated (26-050-070) was purchased from Fisher Scientific. Fetal Bovine Serum (FBS) was purchased from Avantor Seradigm (lot # 214B17). All cell lines were obtained from ATCC and were maintained at a low passage number (<20 passages). HEK293T (ATCC: CRL-3216) cells were cultured in DMEM supplemented with 10% FBS and 1% antibiotics (Penn/Strep, 100 μ/mL). MIA-PaCa-2 (ATCC: CRL-1420) cells were cultured in DMEM supplemented with 10% FBS, 1% antibiotics (Penn/Strep, 100 μ/mL), and 2.5% horse serum. H661 (ATCC: HTB-183), H1437 (ATCC: CRL-5872), H358 (ATCC: CRL-5807), HCT-15 (ATCC: CCL-225), Jurkat (ATCC: TIB-152), MOLT-4 (ATCC: CRL-1582) and H2122 (ATCC: CRL-5985) cells were cultured in RPMI-1640 supplemented with 10% FBS and 1% antibiotics (Penn/Strep, 100 μ/mL). HEC-1-B (ATCC: HTB-113), MeWo (ATCC: HTB-65), and CaCo-2 (ATCC: HTB-37) cells were cultured in EMEM supplemented with 10% FBS and 1% antibiotics (Penn/Strep, 100 μ/mL). Cells were maintained in a humidified incubator at 37 °C with 5% $CO_2$. Cells were harvested by centrifugation (4500 × *g*, 5 min, 4 °C) and washed twice with cold DPBS. Cell pellets were then lysed with sonication (amp = 10, 10 × 1 sec pulses). The lysates were then transferred to a new microcentrifuge tube. Protein concentrations were determined using a BioRad DC protein assay kit from Bio-Rad Life Science (5000113, 5000114) and the lysate diluted to the working concentrations indicated below.

### RNA-seq variant calling

Total RNA was extracted from cells using the Invitrogen Purelink RNeasy Plus Mini Kit (Qiagen, 166043750) or PureLink RNA mini kit (ThermoFisher, 12183018 A) or Library preparation and RNA sequencing was carried out by the UCLA Technology Center for Genomics and

Bioinformatics (TCGB). Libraries were prepared using the KAPA-stranded mRNA kit. Paired-end sequencing (2 × 150) was performed to a depth of 50–60x with an Illumina HiSeq3000 system. RNAFastq paired-end reads for each cell line were aligned to Gencode reference genome hg38 (GRCh38.p13) using STAR-2 PASS alignment (v2.7.3a)[114]. We ran STAR with default settings for paired reads and the following additional parameters:−outSAMtype BAM SortedByCoordinate, --outSAMunmapped Within, and --sjdbFileChrStartEnd in the second alignment. Samtools (v1.7)[115,116] calmd was used to add MD tags to sorted BAM files. Opposum (v0.2, 02-23-2017)[117] was used to split reads and mark duplicates using default parameters and an additional parameter --SoftClipsExist True. Platypus (v0.8.1)[118] was used to call variants and generate VCF files using default parameters. Samtools flag stat was used to obtain BAM file-mapped read counts. Raw reads submitted to Sequence Read Archive (SRA) as BioProject PRJNA997729.

## WE-seq variant calling
Genomic DNA was extracted from cells using the Zymo Quick DNA Miniprep Plus Kit kit (Fisher Sci, 50-444-149). Library preparation and exome sequencing was carried out by the UCLA Technology Center for Genomics and Bioinformatics (TCGB). Libraries were prepared using the Nimblegen Capturing Kit. Paired-end sequencing (2 × 150) was performed to a depth of 50–60x with an Illumina HiSeq3000 system. Fastq paired-end reads for each cell line were aligned to Gencode reference genome hg38 (GRCh38.p13) using BWA-MEM alignment and default parameters for paired reads. Output SAM files were converted to BAM files using Samtools (v1.7) and Picard (v2.21.4) (https://broadinstitute.github.io/picard/) was used to generate coordinate-sorted BAM files with read groups added. Samtools was used to index the files and duplicates were marked with Picard. GATK-HaplotypeCaller (v4.1.8.1)[119] was used to split reads. Since we do not have matched normal samples, we opted to use the germline caller GATK-HaplotypeCaller for exome data. Variants were called using default parameters with the exception of the ploidy option which was set to the value outlined in Supplementary Data 3 (tab 23). GATK was used to index the VCF file and filter the variants using the following parameters: -window 35 -cluster 3 --filter-name FS --filter-expression "FS >30.0" --filter-name QD --filter-expression "QD <2.0". Samtools flag stat was used to obtain BAM file-mapped read counts. Raw reads submitted to Sequence Read Archive (SRA) as BioProject PRJNA997729.

## HCT-15 expression analysis
5 biological replicate RNA extracts were sequenced as described and aligned to hg38 as described in *RNA-seq variant calling*. Kallisto (v0.46.1)[120] was used to estimate transcript counts with indexed Gencode v28 transcriptome (gencode.v28.transcripts.fa) and -b (bootstrap) set to 100. Abundance transcript files were normalized with DEseq2 (v1.28.1)[121]. The Counts table was subsetted to a curated set of nonredundant CCDS transcript ID's (24,950) in Supplementary Data 3 (tab 21), and mean counts were calculated for downstream analysis. Raw reads submitted to Sequence Read Archive (SRA) as PRJNA997729.

## Predicting amino acid changes
Supplementary Data 3 (tab 21) (nonredundant CCDS transcript ID's) was used to remove redundant proteins. VCFs from variant calling pipelines for both RNA and WES were processed using R package 'Variant Annotation'[122]. First, a TxDB object was made using the Gencode v28 annotation GTF file. The 'predictCoding' function using genome hg38 (GRCh38.p13) was used to obtain protein level changes from the VCFs, and 'nonsynonymous' and 'nonsense' changes were extracted; the resulting table includes a set of internal transcript IDs labeled 'TXID'. A database of common SNPs from NCBI (04-23-2018

00-common_all.vcf.gz) was used to annotate SNPs from rare mutations. The output missense table (Supplementary Data 3 tab 22) lists reference/variant codons and amino acids. This table was filtered to contain matches to the CCDS set of 24,950 Ensembl transcript IDs only and those that resulted in single amino acid variants (SAAVs), ignoring small indels and multi-nucleotide variants (48,552 variants). Variants *passing* variant-calling filters were used in Figs. 3, 4 (48,301 variants), and non-PASS variants are included in the proteomics analyses.

## Generation of sample-specific custom databases with all combinations of variants
Several R packages were used in generating custom databases: *VariantAnnotation*[122], *GenomicFeatures*[123], *biomaRt*[124] and *BSgenome.Hsapiens.UCSC.hg38* (10.18129/B9.bioc.BSgenome.Hsapiens.UCSC.hg38). A curated set of CCDS transcript ID's (24,950) (Supplementary Data 3) was used to subset the Gencode v28 protein coding translations FASTA file by Ensembl transcript IDs. These sequences consist of a non-redundant UniProtKB[125] subset of cross-referenced CCDS proteins. Using the previously generated TxDB object from '*Predicting amino acid changes*', and the biomaRt select function, corresponding TXID headers for the protein FASTA file were obtained by selecting 'TXID' with Ensembl transcript ID keys ('TXNAME'). Matching TXIDs from all SAAVs with new protein sequence TXIDs, positions in the corresponding wild-type protein sequences were replaced with the corresponding variant amino acid to generate a list of protein sequences containing only one variant per sequence. Protein sequences containing variants shared between RNA and exome-derived variants were grouped as one sequence. For proteins containing multiple variants, all possible combinations were generated for variants within 30 amino acid windows for proteins with 25 (or 15, see SI tables) or fewer total variants. Output sequences were written to a FASTA file with headers containing corresponding Uniprot-ID, Gene ID, Ensembl transcript ID, and missense changes, as well as cell-line, and sequencing origin (RNA or WE). To limit the increased search space, the database variant protein sequences were *in-silico* digested. A custom Python script was used to *in-silico* digest the FASTA to generate tryptic peptides containing 2 miss cleavages (two tryptic sites flanking amino acids surrounding the individual variant). Any duplicated peptide sequences were removed to leave unique sequences. For compatibility with MSFragger-based searches, simplified FASTA headers were used containing only the Uniprot ID. Result peptides are mapped back to detailed FASTA files for variant information. Scripts are available at https://github.com/BackusLab/chemoproteogenomics.

## Proteomic sample preparation for unenriched sample analysis
HCT-15 and MOLT-4 lysates were incubated in 2 M urea/PBS at RT (final concentration = 2 mg/mL). DTT (10 μL of 200 mM stock in water, final concentration = 10 mM) was added to each sample, and the sample was incubated at 65 °C for 15 min. To this, iodoacetamide (10 μL of 400 mM stock in water, final concentration = 20 mM) was added, and the solutions were incubated for 30 min at 37 °C. Following the addition of 3 μL trypsin solution (Worthington Biochemical, LS003740, 1 mg/mL in 666 μL of 50 mM acetic acid and 334 μL of 100 mM $CaCl_2$, final weight = 2 ng), digest was allowed to proceed overnight at 37 °C with shaking. The next day, 90 μL from each digest was combined with 210 μL water and 0.3 μL TFA (final concentration ~0.1% TFA and ~180 μg peptides). Samples were fractionated into low-bind Eppendorf tubes using a high-pH reversed-phase fractionation kit (Pierce, 84868). Fractions were dried (Speed Vac) and then reconstituted with 15 μL 5% acetonitrile and 1% FA in MB water and analyzed by LC-MS/MS. Samples were fractionated in triplicates for a total of 48 samples.

## Proteomic sample preparation for cysteine-enrichment sample analysis

Proteome samples (200 µL of 1 mg/mL, prepared as described in the preparation of cell lysates) Samples were then labeled with 2 mM IAA (2 µL of 200 mM stock solution in DMSO, final concentration = 2 mM) for 1 h at RT (700 rpm). CuAAC was performed with biotin azide (2) (4 µL of 200 mM stock in DMSO, final concentration = 4 mM), TCEP (4 µL of fresh 50 mM stock in water, final concentration = 1 mM), TBTA (12 µL of 1.7 mM stock in DMSO/tbutanol 1:4, final concentration = 100 µM), and CuSO$_4$ (4 µL of 50 mM stock in water, final concentration = 1 mM) for 1 h at ambient temperature. After CuAAC, 10 µL of 20% SDS was added to each sample. Samples were incubated with 0.5 µL benzonase (Fisher Scientific, 70-664-3) for 30 min at 37 °C. The samples were then subjected to SP3 sample loading, SP3 digest and elution, NeutrAvidin enrichment, and LC-MS/MS analysis, as described below. Experiments were conducted in duplicate for each cell line.

## Proteomic sample preparation for ligandability screening

HCT-15, MOLT-4, and MeWo proteome samples (200 µL of 2 mg/mL, prepared as described in the preparation of cell lysates). Compound (500 µM) or DMSO vehicle was added to lysates for 1 hr (2 µL 50 mM stocks or 2 µL DMSO). Samples were chased with 2 mM IAA (2 µL of 200 mM stock solution in DMSO, final concentration = 2 mM) for 1 hr. CuAAC was performed with heavy biotin azide (DMSO samples) or light biotin azide (compound labeled samples) (4 µL of 200 mM stock in DMSO, final concentration = 4 mM), TCEP (4 µL of fresh 50 mM stock in water, final concentration = 1 mM), TBTA (12 µL of 1.7 mM stock in DMSO/tbutanol 1:4, final concentration = 100 µM), and CuSO4 (4 µL of 50 mM stock in water, final concentration = 1 mM) for 1 h at ambient temperature. After CuAAC, 10 µL of 20% SDS was added to each sample. Samples were incubated with 0.5 µL benzonase (Fisher Scientific, 70-664-3) for 30 min at 37 °C. The samples were then subjected to *SP3 sample loading* using 80 µL total bead volumes, SP3 digest and elution, NeutrAvidin enrichment, and LC-MS/MS analysis, as described below. Experiments were conducted in triplicate for each compound per cell line.

## Proteomic sample preparation with TMT labeling

Jurkat lysates (each at 400 µg) were treated with either DMSO or cysteine reactive compounds (KB2, KB3, JC19) for 1 h at ambient temperature, each in duplicate. Treated lysates are then labeled with 200 µM IAA for 1 h in the dark and then 'clicked' (via CuACC) with biotin azide. CuACC conditions were as follows: (4 µL of 200 mM stock in DMSO, final concentration = 4 mM), TCEP (4 µL of fresh 50 mM stock in water, final concentration = 1 mM), TBTA (12 µL of 1.7 mM stock in DMSO/tbutanol 1:4, final concentration = 100 µM), and CuSO4 (4 µL of 50 mM stock in water, final concentration = 1 mM). The samples were then subjected to *SP3 sample loading* using 80 µL total bead volumes, *SP3 digest and elution*, and *NeutrAvidin enrichment*, as described below. Samples were then quantified using a Pierce™ quantitative peptide concentration assay (ThermoFisher Scientific, 23275). Enriched samples were speed-vacc'd, resuspended in 100 mM TEAB, and labeled with TMT 10plex reagents (ThermoFisher Scientific, 90114) at a ratio of (6:1 µg TMT tag to µg of peptide) for 1 h at ambient temperature followed by quenching with 5% hydroxylamine and acidification with 5% formic acid. Samples were then cleaned up using Pierce™ C18 spin tips (Thermo Fisher, Cat. No. 87784) according to the manufacturer's instructions. Samples were then combined and dried with Speedvac and reconstituted with mass spectrometry solvent (5% acetonitrile and 1% formic acid in molecular-biology grade water) and analyzed by LC-MS/MS.

## SP3 sample loading

SP3 sample cleanup was performed generally at a bead/protein ratio of 10:1 (wt/wt) (38). For each 200 µL sample, 20 µL (or 40 µL) Sera-Mag SpeedBeads Carboxyl Magnetic Beads, hydrophobic (GE Healthcare, 65152105050250, 50 µg/µL, total 1 mg) and 20 µL (or 40 µL) Sera-Mag SpeedBeads Carboxyl Magnetic Beads, hydrophilic (GE Healthcare, 45152105050250, 50 µg/µL, total 1 mg) were aliquoted into a single microcentrifuge tube and gently mixed. Tubes were then placed on a magnetic rack until the beads settled to the tube wall, and the supernatants were removed. The beads were removed from the magnetic rack, reconstituted in 1 mL of MB water, and gently mixed. Tubes were then returned to the magnetic rack, beads allowed to settle, and the supernatants removed. Washes 20 were repeated for two more cycles, and then the beads were reconstituted in 40 µL MB water. The bead slurries were then transferred to the proteome samples and incubated for 10 min at RT with shaking (1000 rpm).

## SP3 digest and elution

Absolute ethanol (400 µL) was added to each sample, and the samples were incubated for 5 min at RT with shaking (1000 rpm). Beads were washed twice with 80% ethanol as described above. Beads were then resuspended in 200 µL 0.5% SDS in PBS containing 2 M urea. DTT (10 µL of 200 mM stock in water, final concentration = 10 mM) was added to each sample, and the sample was incubated at 65 °C for 15 min. To this, iodoacetamide (10 µL of 400 mM stock in water, final concentration = 20 mM) was added, and the solution was incubated for 30 min at 37 °C with shaking. After that, absolute ethanol (400 µL) was added to each sample, and the samples were incubated for 5 min at RT with shaking (1000 rpm). Beads were then again washed three times with 80% ethanol in water (400 µL). Next, beads were resuspended in 150 µL PBS containing 2 M urea, followed by the addition of 3 µL trypsin solution (Worthington Biochemical, LS003740, 1 mg/mL in 666 µL of 50 mM acetic acid and 334 µL of 100 mM CaCl$_2$, final weight = 2 ng). Digest was allowed to proceed overnight at 37 °C with shaking. After digestion, ~4 mL acetonitrile (>95% of the final volume) was added to each sample, and the mixtures were incubated for 10 min at RT with shaking (1000 rpm). Supernatants were then removed and discarded using the magnetic rack, and the beads were washed (3 × 1 mL acetonitrile). Peptides were then eluted from SP3 beads with 100 µL of 2% DMSO in MB water for 1 h at 37 °C with shaking (1000 rpm). The elution was repeated again with 100 µL of 2% DMSO in MB water. A peptide concentration assay (Pierce, 23275) was performed to test the concentration of the peptide. The elution can be used for NeutrAvidin enrichment or analyzed by LC-MS/MS.

## NeutrAvidin enrichment of labeled peptides

For each sample, 50 µL of NeutrAvidin® Agarose resin slurry (Pierce, 29200) was washed three times in 10 mL IAP (immunoaffinity purification) buffer (50 mM MOPS−NaOH (pH 7.2), 10 mM Na$_2$HPO$_4$, 50 mM NaCl) and then resuspended in 500 µL IAP buffer. Peptide solutions eluted from SP3 beads were then transferred to the NeutrAvidin® Agarose resin suspension, and the samples were then rotated for 2 h at RT. After incubation, the beads were pelleted by centrifugation (21,000 × *g*, 1 min) and washed by centrifugation (3 × 1 mL PBS, 6 × 1 mL water). Bound peptides were eluted with 60 µL of 80% acetonitrile in MB water containing 0.1% FA (10 min at RT). The samples were then collected by centrifugation (21,000 × *g*, 1 min), and residual beads were separated from supernatants using Micro BioSpin columns (Bio-Rad). The remaining peptides were then eluted from pelleted beads with 60 µL of 80% acetonitrile in water containing 0.1% FA (10 min, 72 °C). Beads were then separated from the eluants using the same Bio-Spin column. Eluents were collected by centrifugation (21,000 × *g*, 1 min) and dried (SpeedVac). The samples were then reconstituted with 5% acetonitrile and 1% FA in MB water and analyzed by LC-MS/MS.

## Liquid-chromatography tandem mass spectrometry (LC-MS/MS) analysis

The samples were analyzed by liquid chromatography tandem mass spectrometry using a Thermo Scientific™ Orbitrap Eclipse™ Tribrid™ mass spectrometer coupled with a High Field Asymmetric Waveform Ion Mobility Spectrometry (FAIMS) Interface. Peptides were resuspended in 5% formic acid and fractionated online using an 18 cm long, 100 µM inner diameter (ID) fused silica capillary packed in-house with bulk C18 reversed phase resin (particle size, 1.9 µm; pore size, 100 Å; Dr. Maisch GmbH). The 70-min water acetonitrile gradient was delivered using a Thermo Scientific™ EASY-nLC™ 1200 system at different flow rates (Buffer A: water with 3% DMSO and 0.1% formic acid and Buffer B: 80% acetonitrile with 3% DMSO and 0.1% formic acid). The detailed gradient includes 0 – 5 min from 3 % to 10 % at 300 nL/min, 5 – 64 min from 10 % to 50 % at 220 nL/min, and 64 – 70 min from 50 % to 95 % at 250 nL/min buffer B in buffer A. For bulk fractionation data, the detailed 80 min gradient includes 0 – 3 min from 1 % to 10 % at 300 nL/min, 3 – 63 min from 10 % to 40 % at 220 nL/min, 63 – 73 min from 40 % to 50 % at 220 nL/min, and 73 – 80 min from 50 % to 95 % at 250 nL/min buffer B in buffer A. Data was collected with charge exclusion (1, 8, >8). Data was acquired using a Data-Dependent Acquisition (DDA) method comprising a full MS1 scan (Resolution = 120,000) followed by sequential MS2 scans (Resolution = 15,000) to utilize the remainder of the 1 s cycle time. Time between master scans was set 1 s and 3 s for compound labeling datasets, validation datasets, and fractionation datasets. HCD collision energy of MS2 fragmentation was 30 %. Raw file names used for figures are in Supplementary Data 7.

## Command-line MSFragger-based variant peptide identification and quantitation

Raw data collected by LC-MS/MS were searched using a 2-stage search scheme implemented using custom bash scripts: MSFragger (version 3.5), Philosopher (version 4.2.2) and IonQuant (version 1.8.0) enabled[45,46,48,49]. Precursor and fragment mass tolerance was set as 20 ppm. Missed cleavages were allowed up to 2. Peptide length was set to 7 – 50, and peptide mass range was set to 500 – 5000. Cysteine residues were searched with variable modifications at cysteine residues for carboxyamidomethylation (+57.02146), biotin-azide (+463.2366), and heavy biotin-azide (+469.2742) added for quant searches in Fig. 8 datasets. Labeling was set allowing for 3 max occurrences and 'all mods used in first search' checked. Peptide and protein level FDR were set to 1%. For ligandability screening, permissive IonQuant parameters allowed minimum scan/isotope numbers set to 1. First, raw spectra are searched with normal reference protein sequences (CCDS set), and peptide-to-spectrum matches (PSM) are filtered to 1% FDR. Custom bash scripts were used to extract 1% FDR-filtered PSM scan numbers from this first search. Prior to a second search using the same parameters and custom database, a text file of these scan numbers is generated with leading zeros removed and included as option 'excluded_scan_list_file', allowing remaining scans to be searched with a cell line-specific custom database containing Uniprot identifiers and tryptic peptide sequences as described in Generation of sample-specific custom databases. PeptideProphet[126] was used for rescoring for both searches. Bash scripts are available at https://github.com/BackusLab/chemoproteogenomics.

## FragPipe label-free quantitation of HCT-15 fractionation data

Raw data collected by LC-MS/MS were searched with the default LFQ-MBR workflow provided by FragPipe. With each experimental group corresponding to fractionation set for a total of three intensity values per protein in combined.protein.tsv output. Mean LFQ intensities were calculated, and the non-redundant set of UniProt IDs (Supplementary Data 3 (tab 21)) was used in downstream analyses.

The MS search results and fasta files have been deposited to the ProteomeXchange Consortium (http://proteomecentral. proteomexchange.org) via the PRIDE partner repository[127] with the dataset identifiers PXD043879 for newly generated data, and PXD023059 and PXD029500 for re-analyzed data..

## Transient expression of HLA-B alleles

Expression plasmids (pTwist CMV) containing HLA-B*38:01, HLA-B*27:05, HLA-B*38:01 C91S, and HLA-B*27:05 C91S inserts with C-terminal FLAG-tags were obtained from Twist Bioscience. pDONR223_B2M_WT was a gift from Jesse Boehm & William Hahn & David Root (Addgene plasmid # 81810; http://n2t.net/addgene:81810; RRID:Addgene_81810) and subcloned using GateWay cloning into C-terminal FLAG destination vector generated from a pRK5 backbone vector, which was a kind gift from T Wucherpfennig. Plasmids were co-transfected into 60% confluent 6 cm plated 293 T cells using 14 µL PEI, 140 µL serum-free DMEM, and 1 µg co-transfections or 2 µg eGFP expression plasmid. Cells were harvested after 24 h transfections. Construct sequences and sources available in Supplementary Data 8.

## FLAG-IP and Gel based-ABPP of HLA-B alleles: cell surface and lysate labeling

Cells were washed once with PBS and resuspended in 100 µL serum-free DMEM. One-half of the cells were rotated at RT in 200 µM IAA (1 µL of 10 mM IAA stock) for 1 hr for cell-surface labeling. After spinning down at 1800 × g, the supernatant was removed. Cells were lysed in 30 µL 2% CHAPS/PBS for 30 min on ice. Remainder cells were lysed in 30 µL 2% CHAPS/PBS. Dilute all samples to 300 µL with PBS and spin 1800 × g for 1 min. Samples were adjusted to 2 mg/mL using a Bio-Rad DC protein assay kit from Bio-Rad Life Science (Hercules, CA). 200 µL of unlabeled lysates were incubated with 200 µM IAA (2 µL of 20 mM IAA stock) for 1 hr RT. 50 µL EZred FLAG bead suspension per sample (Sigma, F2426) was washed with tris-buffered saline (TBS) buffer according to manufacturer recommendations. 50 µL washed beads were added to each sample and rotated for 2 h at 4 C. Beads were washed 3x with 500 µL TBS pelleted at 8200 xg and resuspended in 50 µL PBS with 250 µg/mL 3x FLAG peptide (Sigma, F4799) and 0.2% NP-40 alternative (Millipore Sigma, 492016) and rotated for 30 min at 4 C. Beads were pelleted at 8200 xg to capture eluted proteins. Eluant was clicked on to rhodamine-azide (Click Chemistry Tools, AZ109-5) (25 µM rhodamine-azide (1.25 mM stock), 1 mM Tris(2carboxyethyl) phosphine (TCEP) (SigmaAldrich) (50 mM stock), 100 µM Tris[(1-benzyl-1H-1,2,3-triazol-4-yl)methyl]amine (TBTA) (Sigma-Aldrich) (1.7 mM stock), 1 mM CuSO4 (50 mM stock)) for 1 h at RT. All samples were denatured (5 min, 95 °C) and loaded onto 4–12% Criterion™ XT Bis-Tris gels with XT MOPS running buffer from Bio-Rad followed by semi-dry transfer to nitrocellulose membrane. 1:2000 dilution of anti-FLAG rabbit antibody (14793, Cell Signaling) and followed by 1:5000 IRDye® 800CW Goat anti-Rabbit IgG (102673-330, VWR) as well as 1:3000 GAPDH rabbit antibody (2118S, Cell Signaling) followed by 1:5000 IRDye® 680RD Goat anti-Rabbit IgG was used for visualization of loading and rho signal for IAA labeling.

## CAND1 Mutagenesis

G1069C and G1069W point mutations were introduced to WT pcDNA3-HA2-CAND1 plasmid (Addgene, plasmid #20719) using site-directed mutagenesis. For G1069C: PCR samples were prepared by mixing water (33 µL), 5X GC buffer (Phusion) (10 µL), DMSO (2 µL), dNTP Mix RTU (Zymo) (5 µM, 1 µL), CAND1 plasmid (299 ng/µL, 1 µL), the given forward and reverse primers (10 µM, 1 µL), and DNA polymerase (Phusion) (1 µL). For G1069W: PCR samples were prepared by mixing water (34 µL), 5X GC buffer (Phusion) (10 µL), DMSO (1 µL), dNTP Mix RTU (Zymo) (5 µM, 1 µL), CAND1 plasmid (299 ng/µL, 1 µL), the given forward and reverse primers (10 uM, 1 µL), and DNA polymerase (Phusion) (1 µL). PCR amplifications were then performed by heating the samples at the following temperatures: Step 1: 98 °C (3 min); Step 2: 25 cycles of 98 °C (10 sec), 56 °C (30 sec), 72 °C (5 min);

Step 3: 72 °C, 5 min. Plasmid amplification was confirmed by DNA agarose gel. The PCR products (25 μL) were digested by the addition of Dpn1 (1 μL) and heated at 37 °C for 3 h before being transformed into competent Stbl3 cells. Stbl3 cells (50 μL) were thawed on ice for 10 mins and then incubated with the PCR products (2 μL) on ice for a further 10 mins. The cells were heated at 42 °C for 45 sec then cooled on ice for 2 min. Room temperature SOC (500 μL) was added, and the cells were shaken (200 rpm) at 37 °C for 1 h. The mixtures (30 μL) were spread over agar plates containing ampicillin and then incubated at 37 °C overnight. Single colonies were picked and added to LB (5 mL) with ampicillin (100 μg/mL) then the mixtures were shaken (200 rpm) at 37 °C overnight. Plasmids were purified using a Zyppy Plasmid Miniprep Kit (Zymo) and successful mutagenesis was confirmed by DNA sequencing. Construct sequences and sources available in Supplementary Data 8.

## Cul1 and CAND1 Co-Transfections and FLAG-IP

HEK293T cells were co-transfected with 3xFLAG-CUL1-pCMV7.1 (Addgene, plasmid #155019) along with the given CAND1 plasmid (WT, G1069C, G1069W) or control FLAG-GFP plasmid. To the mixed plasmids (5 μg total DNA) was added serum-free DMEM (350 μL) followed by PEI MAX- Transfection Grade Linear 39 Polyethylenimine Hydrochloride (MW 40,000) (Polysciences, Inc., 24765-1) (35 μL of 1 mg/mL, pH 7.4). The solutions were mixed and incubated for 15 min at room temperature before being added dropwise to 10 cm dishes at 60–70% cell confluency. The cells were incubated at 37 °C, 5% $CO_2$ in a humidified incubator for ~44 h before being harvested. The cells were washed with cold phosphate-buffered saline (PBS) (2 × 3 mL) by resuspension and centrifugation (300 × $g$, 3 min, 4 °C). The washed cells were collected into cold PBS (1 mL), centrifuged (300 × $g$, 3 min, 4 °C), and the supernatant was discarded. The transfected cell pellets were lysed by suspending in a cold solution of 1% NP-40, 50 mM HEPES (pH 7.4), 150 mM NaCl, cOmplete™ Mini EDTA-free Protease Inhibitor Cocktail (1 tablet per 10 mL) (Roche) (400 μL) and incubating on ice for 30 min. The lysates were centrifuged (15000 × $g$, 5 min, 4 °C) and the soluble fractions were collected. Protein concentrations were measured using a DC Protein Assay Kit (BioRad) (5000113, 5000114) and each lysate was diluted to 2 – 3 mg/mL in the lysis buffer, with all samples from the same experiment normalized to the same concentration. EZview™ Red ANTI-FLAG® M2 Affinity Gel resin (MilliporeSigma) (13 μL per sample) was washed with cold Tris-Buffered Saline (TBS) (2 × 1 mL) by suspension and centrifugation (8200 × $g$, 30 sec, 4 °C) before being suspended in cold TBS (100 μL per sample) and distributed to each lysate sample. The samples were tumbled at 4 °C for 2 h before being centrifuged (8200 × $g$, 30 sec, 4 °C) and the supernatant was decanted. The resin was washed with cold TBS (3 × 600 μL) by resuspension and centrifugation (8200 × $g$, 30 sec, 4 °C) before being resuspended in 4X Laemmli buffer (BioRad) (40 μL) and heated at 95 °C for 5 min. Lysate input and flowthrough samples were prepared by mixing lysate or supernatant (50 μL) with 4X Laemmli buffer with 10% β-mercaptoethanol (17 μL) and then heating at 95 °C for 5 min. The samples were loaded onto 26 well 4–20% Criterion™ TGX Stain-Free™ Protein Gels (BioRad) along with Precision Plus Protein™ All Blue Prestained Protein Standards ladder (BioRad) and protein bands were separated at 140 V. The gels were transferred to nitrocellulose (0.2 μM) membranes (BioRad) using a Trans-Blot Turbo System (BioRad) with the 'Mixed MW' setting. The transferred membranes were blocked in 5% (w/v) milk in TBS for either 1 h at room temperature or overnight at 4 °C. The membranes were washed with TBS (3 × 5 min), cut to separate the relevant high and low molecular weight bands, then incubated with 1:1000 (CUL1 Rb Antibody #4995, Cell Signaling; anti-HA-Tag Rb mAb #C29F4, Cell Signaling; anti DDDDK-Tag Rb mAb #D6W58, Cell Signaling) or 1:2000 (β-Actin Rb mAb #AC038, ABclonal) primary antibody in 5% (w/v) milk in TBS at 4 °C overnight. The membranes were washed with TBS (3 × 5 min) and then incubated with 1:3000 IRDye 800CW Goat anti-Rabbit IgG Secondary Antibody (LI-COR) in 5% (w/v) milk in TBS with 0.1% Tween20 (Fisher) for 1 h at room temperature. The membranes were washed with TBS (3 × 5 min) and imaged using a ChemiDoc Imaging System. Construct sequences and sources available in Supplementary Data 8.

## HMGB1 expression and purification

Expression constructs obtained from TWIST Bio in pET29b were transformed into *E. coli* BL21-Gold (DE3) cells (Agilent). An overnight culture was used to inoculate 3 L of Terrific broth media supplemented with kanamycin (50 μg/mL). The cultures were grown to an OD600 of 1.0, the temperature shifted to 20 °C, and protein expression induced by the addition of IPTG (final concentration of 0.5 mM). Cell growth was continued overnight, and the cells were harvested by centrifugation. Cell pellets were resuspended in Buffer A (50 mM Tris pH 7.5, 500 mM NaCl, 20 mM imidazole, 3 mM B-ME) supplemented with Complete protease inhibitor (Roche), PMSF (final concentration of 1 mM), EDTA (final concentration of 1 mM), and benzonase. The suspension was lysed with three passes through an Emulsiflex C-3 (Avestin) at 15000 psi. MgCl₂ was added to 1 mM, and the lysate was clarified by centrifugation (25000 × $g$, 30 min, 4 °C). The clarified supernatant was loaded on a 5 ml NiNTA FF Crude column (Cytiva) equilibrated in Buffer A. The column was washed extensively with Buffer A and bound protein eluted with a linear gradient to 100% Buffer B (Buffer A with 300 mM imidazole). Fractions containing HMGB1 mutants were pooled, concentrated using an Amicon Ultra-15 centrifugal concentrator, and further purified by size exclusion chromatography using a Superdex 75 16/600 column (Cytiva) equilibrated in Buffer C (50 mM Tris pH 7.5, 125 mM NaCl, 1 mM DTT, 10% glycerol). Fractions containing HMGB1 mutants were pooled and diluted 1 in 3 with Buffer D (50 mM Tris pH 7.5, 10% glycerol) and loaded on a 1 ml CaptoQ column (Cytiva) equilibrated in Buffer E (50 mM Tris pH 7.5, 50 mM NaCl, 1 mM DTT, 10% glycerol). After washing, bound protein was eluted with a gradient to 100% Buffer F (Buffer E with 1 M NaCl). Fractions containing the target protein were concentrated with Amicon Ultra-15 centrifugal concentrators. The protein was aliquoted, flash-frozen with liquid nitrogen, and stored at − 80 °C pending assaying.

## HEK-Blue pattern recognition receptor (PRR) activation assay and analysis

hTLR4 and Null control HEK-Blue reporter cell lines (InvivoGen, San Diego, CA) were maintained and used in PRR assay according to previously published protocols[76] and according to the manufacturer's instructions. For PRR assay, 20 μL of HMGB1 proteins (commercially manufactured by Tecan OR lab-manufactured) or 20 μL of positive control LPS were added to respective wells of 96-well flat bottom plates at various concentrations followed by 180 μL of either hTLR4 or Null HEK-Blue reporter cells in HEK-Blue Detection media (InvivoGen). Final working concentrations were as follows: all-thiol HMGB1 (0.2 μg/ML, Tecan); di-sulfide HMGB1 (0.2 μg/mL, Tecan); WT HMGB1, R110C, and R110W mutant (0.2 μg/mL, 0.6 μg/mL, 1 μg/mL for each). Plates were incubated at 37 °C, 5% $CO_2$ for 16 h, and SEAP activity was measured by reading optical density (OD) at 650 nm with a SpectraMax M2 microplate reader (Molecular Devices, Sunnyvale, CA). To determine response ratios, sample OD values (normalized with media-only background absorbance subtracted) were divided by normalized "cells + $H_2O$" control OD values. Non-linear regression was used to plot and analyze response ratio curves and to obtain $EC_{50}$ and $EC_{max}$ values of positive control LPS for hTLR4 cells and Null controls. The final working concentrations of LPS were as follows: 1 μg/mL, 0.1 μg/mL, 0.01 μg/mL, 0.001 μg/mL, 0.0001 μg/mL, and 0.00001 μg/mL. Prism 10.1 (GraphPad Software, La Jolla, CA) was used to plot data and for statistical analysis. Statistical significance was determined via unpaired student's t-test, and a $p$-value of <0.05 was considered significant.

## Data processing for cys-enrichment and quantitative ratio peptide analysis

Custom R scripts were implemented to compile modified_peptide_label_quant.tsv (quant) outputs from the command line MSFragger pipeline or FragPipe to count unique quantified cysteines. Unique cysteines were quantified for each dataset using unique identifiers consisting of a UniProt protein ID, the amino acid number of the modified cysteine, and an additional parameter specifying single or double isotopic labeling (heavy and/or light). Unique proteins were established based on UniProt protein IDs. Residue numbers were found by aligning the peptide sequence to the corresponding UniProt ID protein sequence specified by FragPipe outputs. Variant residue sites were obtained by mapping peptide sequences to respective custom FASTA files containing variant info in the headers. For enriched samples, nonspecific non-Cys-containing peptides were omitted from the analysis. For ratios data, methionine oxidized peptides were omitted, and unpaired heavy or light-identified peptides were kept by setting ratios to $\log_2(20)$ or $\log_2(1/20)$. Outputs were generated by taking the median H:L ratio among all tryptic peptides for unique cysteines in replicate datasets (modified_peptide_label_quant.tsv); mean ratio values were calculated across replicate datasets; quantified cysteines appearing in at least two replicates with ratio SD = <1 were kept (Figs. 8, 6 data) and no SD filter was applied for Fig. 1 data to interpret ratio skew. For comparisons to CysDB cysteines, unique UniprotID_CysPosition identifiers were used and FragPipe assigned Protein ID was used; for ClinVar variant matching, chromosome position and nucleotide changes of associated variants were used. The MS search results and FASTA files have been deposited to the ProteomeXchange Consortium via the PRIDE partner repository with the dataset identifiers PXD043879 for newly generated data and FASTA files. R and Python scripts are available at https://github.com/BackusLab/chemoproteogenomics

## Data processing for high-pH data analysis

Custom R scripts were implemented to compile peptide.tsv outputs from command line MSFragger pipelines. Unique proteins were established based on UniProt protein IDs. Variant residue sites were obtained by mapping peptide sequences to respective custom FASTA files containing variant info in the headers. Residue numbers were found by aligning the peptide sequence to the corresponding UniProt ID protein sequence specified by FragPipe outputs. For ClinVar variant matching, chromosome position and nucleotide changes of associated variants were used.

## Linear sequence and spatial site-analysis

For linear sequence analysis in Fig. 6 datasets, residues 'in or near' UniProtKB annotated active or binding sites, DisProt annotated sites[128–130], or PhosphoSite annotated sites[81] were within +/− 10 amino acids in linear sequence. For Fig. 8 datasets, residues 'in or near' UniProtKB annotated active or binding sites, disprot annotated sites, or phosphosite annotated sites were assessed using 3D Protein Data Bank (PDB) structures. PDB structures were parsed to find all neighboring residues within a 10 Angstrom distance of the liganded Cys (alpha carbon atom). PDB_UniProtKB identifiers were created for each cysteine and a corresponding list of neighboring residues. If the UniProtKB annotated active or binding sites were resolved in an associated crystal structure and found within the 10 Angstroms net, they were classified as cysteine proximal to a known active or binding site.

## Reporting summary

Further information on research design is available in the Nature Portfolio Reporting Summary linked to this article.

## Data availability

The MS search results and fasta files have been deposited to the ProteomeXchange Consortium (http://proteomecentral.proteomexchange.org) via the PRIDE partner repository with the dataset identifiers PXD043879 for data generated for this study, and PXD023059 and PXD029500 for re-analyzed datasets. Sequencing data is deposited in the Sequence Read Archive (SRA) as BioProject PRJNA997729. Source data is provided as source data files for Figures and within the Supplementary Information file for Supplementary Figs. Publicly available databases used are COSMIC v96 (https://cancer.sanger.ac.uk/cell_lines), dbSNP database (4-23-18) (https://www.ncbi.nlm.nih.gov/snp/), ClinVar (09-03-22) (https://www.ncbi.nlm.nih.gov/clinvar/), The UniProt Consortium (https://www.uniprot.org/), DisProt (https://disprot.org/), Phosphosite (https://www.phosphosite.org/homeAction.action), Protein Data Bank (https://www.rcsb.org/). Source data are provided with this paper.

## Code availability

MSFragger can be downloaded as a single JAR binary file at https://msfragger.nesvilab.org/. FragPipe is available on GitHub at https://github.com/Nesvilab/FragPipe. Custom code is available at https://github.com/BackusLab/chemoproteogenomics with https://doi.org/10.5281/zenodo.13788083.

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

## Acknowledgements

We thank all members of the Backus lab for their helpful suggestions. We thank the UCLA Technology Center for Genomics and Bioinformatics (TCGB). In addition, we thank Jigar Desai for guidance on NGS data processing, Angela Wei for guidance on Kallisto data processing, and Ian Ford for providing a CuAAC-compatible IP protocol. The results here are in part based upon data generated by the COSMIC-CLP: https://cancer.sanger.ac.uk/cell_lines and TCGA Research Network: https://www.cancer.gov/tcga. This study was supported by a Beckman Young Investigator Award (K.M.B.), V Scholar Award V2019-017 (K.M.B.), UCLA Jonsson Comprehensive Cancer Center Seed Grant (K. M. B.), UCLA DOE Institute DE-FC02-02ER63421 (K.M.B and M.A.), the National Institutes of Health grants R01-GM094231 and U24-CA271037 (A.I.N.), National Institutes of Health KUH-ART TL1 Fellowship fund 5TL1DK132768 (K.V.B), and National Institutes of Health UCLA Chemistry Biology Interface T32GM136614 (M.V.). The content is solely the responsibility of the authors and does not necessarily represent the official views of the National Institutes of Health.

## Author contributions

H.S.D., K.M.B., and A.I.N. conceptualization. H.S.D. formal analysis. H.S.D., K.H.A., K.V.B., and F.S. visualization. H.S.D. validation. H.S.D., L.M.B., F. Y., and K.M.B. data curation. H.S.D., S.O., F.S., K.H.A., K.V.B., M.A.A., and M.V. investigation. H.S.D., F.Y., and N.U. methodology. H.S.D. and K.M.B. writing–original draft. H.S.D., K.H.A., S.O., L.M.B., F.Y., A.I.N., and K.M.B. writing–review and editing. A.I.N., E.F.R., and K.M.B. supervision. A.I.N. and K.M.B. funding acquisition.

## Competing interests

K.M.B. is a member of the advisory board at Matchpoint Therapeutics. A.I.N. and F.Y. receive royalties from the University of Michigan for the sale of MSFragger and IonQuant software licenses to commercial entities. All license transactions are managed by the University of Michigan Innovation Partnerships office, and all proceeds are subject to university technology transfer policy. The remaining authors declare no competing interests.
