## [Peer Review file · Nature Communications]

Chemoproteogenomic stratification of the missense variant cysteinome

Corresponding Author: Professor Keriann Backus

Version 0:

Reviewer comments:

Reviewer #1

(Remarks to the Author)

Key Results: The authors demonstrate the utility of a pipeline combining chemoproteomics with proteogenomics. Specifically, they combine cysteine-containing protein enrichment via IAA-alkyne labeling of cysteines followed by click azide-alkyne conjugation of isotopically-labeled biotin. They show that gain-of-cysteine mutations are common features of both disease and healthy genomes, and that cysteine is one of the most commonly gained amino acids due to missense mutation. Generation of sample-specific databases incorporating whole exome sequencing and RNA-seq data, combined with a strategy for the combinatorial generation of a multi-variant peptide database enables deep characterization of gain-of-cysteine variants and cysteine-proximal variants. This platform enabled deeper coverage of gain-of-cysteine variants in selected high variant burden cell lines than bulk proteogenomics with offline high-pH fractionation. They then show that the presented chemoproteogenomics pipeline effectively facilitates cysteine ligandability screening, and enables the determination of how specific SAAVs proximal to cysteines impact that cysteine's ligandability.

Validity: The authors effectively support their interpretations of results and their conclusions with data and references.

Significance: The chemoproteogenomic platform outlined in this manuscript represents a powerful strategy for identifying and characterizing SAAVs. The authors also demonstrate the utility of various databases (COSMIC, TCGA, UniProt, etc.) and scoring metrics (CADD) for assessment of the importance of specific SAAVs. I believe that this work is of high potential significance in the area of variant protein characterization and in the development of specific variant-targeted therapeutics.

Data and methodology: The data presented appears to be high-quality.

Analytical Approach: The analysis of data and statistical analyses performed are valid and robust.

Suggested Improvements:

- While the work and results presented are high-quality, and certainly of potential interest to a wide variety of fields, the manuscript is complicated by a lack of consistent focus. There appear to be two related but separate narratives going on throughout the manuscript. The first is the development and validation of a novel chemoproteogenomics platform applied to the identification of gain-of-cysteine variants and variants that are proximal to both reference and variant cysteines, and the capabilities of the platform. The second is a comparative analysis of the missense mutational profiles of selected high variant burden cell lines, with a focus on gain-of-cysteine mutations, and the biological insights that can be gleaned therein. While these two narratives are certainly related, and both are interesting and of potential significance, combining them in a single manuscript results in an unfocused and amorphous manuscript and dilutes the impact of the work presented.

- Reference to Table S2 on line 269 (dMMR cell lines are enriched for rare predicted missense changes, including acquired cysteines—subsection) appears to be incorrect. I found the information in Table S13. Please ensure correct SI references.

References: The authors do a good job of supporting their statements with references to previous literature.

Conclusion: The work presented in this manuscript is of publication quality and potential significance. However, it is held back by the lack of a clear guiding narrative. This results in it being easy to lose track of the point of the paper while reading, making the work presented less impactful than it would be otherwise. I believe that the authors would be better served by submitting the work presented here as 2 separate manuscripts such that each can be stronger and more focused than the current manuscript. If this manuscript is to be published, I believe there needs to be improvement in the focus to make this high-quality and important work digestible for readers.

Reviewer #2

(Remarks to the Author)

In this manuscript from the Backus lab, the authors combine genomics with LC-MS chemoproteomics to establish a platform described as chemoproteogenomics for assessing the missense variant cysteinome. Some of the findings from the studies include a genomic stratification of the predicted pathogenicity of acquired cysteines, custom 2-stage database searching (FragPipe), custom combinatorial search datasets (cell line paired), and identifying redox-sensitive and ligandable genetic variants globally. The manuscript is certainly rich with datasets, but I am uncertain the findings address the functional aspects of single amino acid variants (SAAVs) encoded by missense mutations that result in acquisition of cysteine. The manuscript in its current form reads as a proteomics-focused analytical chemistry study that does not have the conceptual or biological advance needed to warrant publication in Nature Communication. The data and findings are important and would be best suited for a mass spectrometry-focused journal. Here are my major critiques:

1. There is a large effort to mine publicly available and in-house generated RNA-seq and chemoproteomic datasets to make interesting observations and correlations. However, the majority of the manuscript is essentially mining one dataset after another to make correlations with very little follow-up on the functional relevance of identified sites. For example, the authors use their chemoproteogenomics approach to discover new variants (20 to be exact) that were not identified in COSMIC, CCLE or ClinVar. The highlight several of these variants in the results (HMGB1 R110C and SARS R302H) with hints towards function and move on to the next list of variants identified by chemoproteogenomics. The ligandability studies identified fragments that bind particular sites on protein variants (KB02 ligand for ALDOA Cys178 (G196G), HMGB1 Cys106 (R110C), HLA-B/C Cys125 (V127L/S123Y)), which sets up at the very least mutagenesis and biochemical binding assays and additional functional assays to elevate the impact of these findings. This is a recurring theme in the manuscript (providing a list of proteins/sites with little to no verification/follow up), which would be appropriate for a proteomics-focused journal but not for a broad audience.

2. I also find the results section reads more like a methods section, which again goes back to point #1 above that the authors appear to be providing lists of protein, sites, variants, and correlations with existing and publicly available datasets with very little follow up. I appreciate the details on mass spectrometry methods, but this level of methodological information makes it difficult to assess the biochemical and biological findings that I would think is the utility of this global approach. The results section ends with a methods like description of developing software to improve sensitivity of variant peptide identification. Again, the manuscript seems to be more appropriate for a proteomics-focused journal.

3. The cysteine profiling studies on HLA do not fit with the rest of the story presented. The majority of the manuscript has emphasized missense variants that result in an acquired cysteine in the context of cancer (KRAS G12C is used frequently as a model protein through the paper). The author continues this theme and perform ligandability screening of several cancer cell lines followed by a rather abrupt change to efforts for expanding HLA cysteine peptide coverage.

Additional comments:

1. The results section in general is very information rich. For example, the authors list metrics from chemoproteomics analysis (page 21 and 27). Are all these data necessary? I find the results section is difficult to read at times and this could be mitigated by reducing the sheer amount of details and focus on the interpretation and impact of the findings.

2. Several of the figures are extremely dense and complicate (Figures 2, 4, 5, 6). Can these figures be simplified so that the main claims/findings are clear?

3. The supporting information has 40+ figures. Are all these necessary?

Version 1:

Reviewer comments:

Reviewer #2

(Remarks to the Author)

I appreciate the authors addition of new functional data presented in Figure 6. While providing initial evidence, I don't find the selection of tryptophan mutations to mimic small molecule binding at gain-of-cysteine mutations convincing. Are the tryptophan point mutations specifically testing cysteine variant function or perhaps the tryptophan mutation is resulting in general misfolding of the protein? If the authors were to select sites not identified as gain-of-cysteine variants on CAND1 or HMGB1 would the resulting mutant proteins show comparable disruption of function? If so, it will be difficult to assign specific functionality to the cysteine variant using this approach without further mechanistic evaluation. The current data supports this observation. For CAND1, the G1069C mutation blocks protein-protein interactions with CUL1 while G1069W mutation rescues this effect. For HMGB1, both R110C and R110W mutants show a decreased TLR4 response. I believe the authors data support more complicated mechanisms with the cysteine variants that is difficult to generalize with a simple tryptophan mutation.

While the authors have substantially revised the manuscript text, I don't believe it is sufficient to address my previous major critique that this manuscript is more appropriate for a proteomics-focused analytical chemistry journal.

Reviewer #3

(Remarks to the Author)

This manuscript describes a chemoproteogenomic workflow to identify gain-of-cysteine mutations and mutations proximal to reference cysteines within a panel of cell lines with varying levels of missense mutation burden. The workflow utilizes RNA-Seq data to generate cell line-specific variant databases that are then matched in a 2-stage sequence to identify peptides with acquired cysteines, or mutations proximal to known cysteines. This workflow was applied to a panel of 11 cell lines with varying levels of mutation burden. RNA and exome sequencing as well as chemoproteomics were applied in this panel of cell lines to identify 59 gained cysteines and 302 cysteine-proximal mutations. Ligandability screening was performed in 3 cell lines using a small panel of electrophilic fragments, identifying 27 liganded cysteines. Two liganding sites were further characterized using C to W mutations to mimic a liganding event. Introduction of C to CAND1 blocked interaction with CUL1, and this was reversed by the W mutation. Introduction of C and W to HMGB1 resulted in reduced response to TLR4.

The authors have addressed many of the comments in the prior round of review, and have restructured their manuscript to improve the general readability. Importantly, they have now included follow-up characterization on the liganding of CAND1 and HMGB1, which is important to demonstrate the functional relevance of the cysteine mutations, and addresses some of the concerns of reviewer 2 regarding the proteomics-only nature of the original submission. Given the general responsiveness to prior comments, I recommend publication of this manuscript in Nature Communications in its current form.

Reviewer #4

(Remarks to the Author)

In their manuscript, Desai et al. expand their chemoproteomics platform to include missense variant Cys residues in addition to canonical Cys. Though certain high-value Cys variants are well known (e.g. KRAS G12C), it turns out that such variants are relatively common in at least some cancers (and cell lines) and could present promising therapeutic targets, though until now they've been invisible to chemoproteomics screens. Here the authors first describe some changes they've made to MSFragger/FragPipe (software for analyzing MS data) to identify these variant peptides. They then go over a series of bioinformatics analyses they undertook to understand the prevalence of missense variants and to select cell lines with high levels of missense variants suitable for targeting missense Cys (and other missense variants near canonical reactive Cys) in their screens. Next, they use chemoproteomics to identify such sites and perform additional informatic analyses to examine the factors that influence which sites they detect. Along the way, they follow up on a couple of specific examples.

Overall, I think this manuscript addresses an important and interesting problem. The work they present is well done and enhances our understanding of these Cys missense variants. Though I didn't see the first draft of this manuscript, it's apparent from the reviews and rebuttal that it underwent extensive revisions. I think the revised version is clear and well written and does a good job of balancing technical details with their broader biological/chemical message.

The editor asked me specifically to comment on the points raised by Reviewer #2. It seems to me that the revisions made by the authors have gone a long way toward addressing these issues. I can see where the reviewer's concern about the technical nature of the work is coming from, for example given the discussion of database searching strategies. Having said that, I think those details are essential to the work, and in my opinion the authors do a good job in this revision of describing that work in a succinct and accessible way. Another concern raised by Reviewer #2 was that there isn't enough follow-up and the various bioinformatic/chemoproteomic analyses aren't well integrated. In its current presentation, I think the chemoproteomics data are clearly presented while the data mining efforts provide important context. And the follow-up work targeting HMGB1 and CAND1 is a significant enhancement. Overall, I do think this paper warrants publication in Nature Communications. Chemoproteomics is inherently interdisciplinary, and I think the authors' work integrating chemoproteomics with proteogenomics to enable drug discovery is conceptually novel enough to warrant presentation to a broad audience – not just proteomics specialists.

One additional comment: the key question in this paper is why the authors detect few missense variants in their chemoproteomics datasets, relative to the number they detect via sequencing. The authors already consider a variety of factors, but I think there's at least one more worth considering. Another biologically intriguing possibility is that these missense mutations are hard to detect in protein form because they tend to make the proteins less stable and prone to degradation, and thus don't persist as long as their wildtype counterparts. The comparison the authors do with overall protein abundance levels doesn't fully account for this possibility because each mutation will only affect one copy of each gene, and the cell lines likely have another normal copy and express a mixture of mutant and wildtype protein. In cases where the mutated form is degraded preferentially, there may still be enough copies of the wildtype form around to maintain overall protein levels, or at least minimize their change. There are already some hints in the data that this could be going on. For example, in line 367 the authors note that "nearly all of the 77 variants in Clinvar and identified by chemoproteomics are annotated as benign". One explanation for this could be that deleterious variants tend to disrupt protein function in ways that lead to their premature degradation and prevent their detection. Along these lines, around line 360 you compare the frequency of deleterious mutations among various subsets of your chemoproteomics data. I'd be curious to know how the frequency of predicted deleterious mutations compares between the SAAV's you detect and the ones you do not detect. Experimentally, you could even test this by looking at mutant peptide abundances (or comparing mutant and non-mutant sequence pairs) following proteasome inhibition, though that may be beyond the scope of this paper.

Another issue the authors consider is the effects of variants on cys oxidation. Around line 435 the authors look at effects of missense mutations on nearby reactive Cys residues and note that they can alter oxidation state in some cases. A related potential reason why they don't detect many missense mutations that introduce new Cys residues could be that while evolution tends to place reactive Cys in chemical environments that favor maintaining their reactive state, these missense

Cys arise sporadically and are more likely to occupy comparatively inhospitable environments, leaving them prone to oxidation or otherwise unreactive and thus not detectable via chemoproteomics. This probably isn't straightforward to address experimentally, but is worth mentioning in the text.

We thank the reviewers for their thorough assessment of our manuscript and for their thoughtful comments and suggestions. We are delighted by the positive endorsement of our manuscript by both reviewers, and we very much appreciate their endorsement that our work is “**high-quality and important,**” “**publication quality and potential significance,**” and that our “**findings are important.**” As described in the point-by-point response, we have incorporated the reviewer feedback into the enclosed revised version. In particular, we are pleased to share that we have substantially reworked the narrative of our resubmission to improve the focus, logical flow, and reader accessibility. Notably, we reorganized the content to address the two-stories-in-one feel and of the first submission and have reworked the figures to remove extraneous content. We also removed substantial methodological details from the results section to enhance clarity and readability. To further broaden the utility of our approach, we have additionally added compatibility with isobaric reagents to our FragPipe two-stage search functionality. Given the widespread interest in proteoform discovery and multi-omics for both functional biology and chemical probe/drug discovery efforts, we expect that these additions should prove widely impactful. We are also particularly pleased to share that we have added functional analysis for variants in HMGB1 and CAND1, which reveal substantial changes in cytokine activity, and protein interactions, respectively. Taken together, we believe that these changes substantially strengthen our manuscript, and we are hopeful the reviewers will assess our study as a valuable addition to the field of proteogenomics, with widespread applications in proteoform discovery and multi-omics for basic and translational applications.

REVIEWER COMMENTS

Reviewer #1 (Remarks to the Author):

Key Results: The authors demonstrate the utility of a pipeline combining chemoproteomics with proteogenomics. Specifically, they combine cysteine-containing protein enrichment via IAA-alkyne labeling of cysteines followed by click azide-alkyne conjugation of isotopically-labeled biotin. They show that gain-of-cysteine mutations are common features of both disease and healthy genomes, and that cysteine is one of the most commonly gained amino acids due to missense mutation. Generation of sample-specific databases incorporating whole exome sequencing and RNA-seq data, combined with a strategy for the combinatorial generation of a multi-variant peptide database enables deep characterization of gain-of-cysteine variants and cysteine-proximal variants. This platform enabled deeper coverage of gain-of-cysteine variants in selected high variant burden cell lines than bulk proteogenomics with offline high-pH fractionation. They then show that the presented chemoproteogenomics pipeline effectively facilitates cysteine ligandability screening, and enables the determination of how specific SAAVs proximal to cysteines impact that cysteine’s ligandability.

Validity: The authors effectively support their interpretations of results and their conclusions with data and references.

Significance: The chemoproteogenomic platform outlined in this manuscript represents a powerful strategy for identifying and characterizing SAAVs. The authors also demonstrate

the utility of various databases (COSMIC, TCGA, UniProt, etc.) and scoring metrics (CADD) for assessment of the importance of specific SAAVs. I believe that this work is of high potential significance in the area of variant protein characterization and in the development of specific variant-targeted therapeutics.

Data and methodology: The data presented appears to be high-quality.

Analytical Approach: The analysis of data and statistical analyses performed are valid and robust.

We thank this reviewer for their enthusiastic assessment of our manuscript and for their thorough and constructive critique. In the point-by-point response below, we have incorporated this reviewer's suggestions, which we believe substantially strengthen the manuscript and make it suitable for publication.

Suggested Improvements:

- While the work and results presented are high-quality, and certainly of potential interest to a wide variety of fields, the manuscript is complicated by a lack of consistent focus. There appear to be two related but separate narratives going on throughout the manuscript. The first is the development and validation of a novel chemoproteogenomics platform applied to the identification of gain-of-cysteine variants and variants that are proximal to both reference and variant cysteines, and the capabilities of the platform. The second is a comparative analysis of the missense mutational profiles of selected high variant burden cell lines, with a focus on gain-of-cysteine mutations, and the biological insights that can be gleaned therein. While these two narratives are certainly related, and both are interesting and of potential significance, combining them in a single manuscript results in an unfocused and amorphous manuscript and dilutes the impact of the work presented.

We are so appreciative of this reviewer's suggestion, as we acknowledge that the somewhat unfocused narrative in our initial submission did detract from the broad accessibility and impact of our findings. We concur with this reviewer that the original writing of this manuscript had some characteristics of a two-in-one story, first our reanalysis of publicly available databases and second our proteogenomic work. We have substantially reorganized this manuscript to address this important point. Modifications include beginning the results section with establishing our proteogenomic platform (Previously **Figure 3**), followed by selection of our cell line panel (Previously **Figure 1**), multi-omic analysis (integrated version of **Figure 2 and 3**). We additionally removed the semi-automated search workflow (previously **Figure 8**) to the supporting information, which improves the coherence of the manuscript. Throughout, we have also improved the logic and connectivity by providing additional interpretation of key results.

- Reference to Table S2 on line 269 (dMMR cell lines are enriched for rare predicted missense changes, including acquired cysteines—subsection) appears to be incorrect. I found the information in Table S13. Please ensure correct SI references.

We thank the reviewer for pointing out this incorrect citation, which has been addressed in the resubmission (now tab D13 of table S2). We have additionally checked to ensure that references to figures and supplementary material are correct.

References: The authors do a good job of supporting their statements with references to previous literature.

We have not substantially changed the references from the previous submission.

Conclusion: The work presented in this manuscript is of publication quality and potential significance. However, it is held back by the lack of a clear guiding narrative. This results in it being easy to lose track of the point of the paper while reading, making the work presented less impactful than it would be otherwise. I believe that the authors would be better served by submitting the work presented here as 2 separate manuscripts such that each can be stronger and more focused than the current manuscript. If this manuscript is to be published, I believe there needs to be improvement in the focus to make this high-quality and important work digestible for readers.

We thank this reviewer for pointing out the presence of multiple parallel narratives in our first submission. We do respectfully disagree that this would be better suited to two separate manuscripts, given the value that the publicly available dataset integration brings to interpretation of our newly generated proteogenomic data. As noted above, to improve the logical flow of this manuscript, we have substantially reorganized the text, removed extraneous content, and added additional subheadings to facilitate the logical flow. We have additionally added new functional studies for variants in CAND1 and HMGB1 (now in **Figure 6H-J**), which we believe further strengthens our manuscript. We hope that this reviewer will concur with us that our newly revised manuscript unites the genomics and proteomics aspects of our prior submission in an improved manner, suitable for publication as one coherent narrative.

Reviewer #2 (Remarks to the Author):

In this manuscript from the Backus lab, the authors combine genomics with LC-MS chemoproteomics to establish a platform described as chemoproteogenomics for assessing the missense variant cysteinome. Some of the findings from the studies include a genomic stratification of the predicted pathogenicity of acquired cysteines, custom 2-stage database searching (FragPipe), custom combinatorial search datasets (cell line paired), and identifying

redox-sensitive and ligandable genetic variants globally. The manuscript is certainly rich with datasets, but I am uncertain the findings address the functional aspects of single amino acid variants (SAAVs) encoded by missense mutations that result in acquisition of cysteine. The manuscript in its current form reads as a proteomics-focused analytical chemistry study that does not have the conceptual or biological advance needed to warrant publication in Nature Communication. The data and findings are important and would be best suited for a mass spectrometry-focused journal. Here are my major critiques:

We thank this reviewer for their careful assessment of our manuscript. We are pleased by their endorsement of the richness of our datasets. As described in our point-by-point response below, we have incorporated all of this reviewer's comments into our revised submission, which we believe substantially strengthens the manuscript. Given the widespread applications and interdisciplinary nature of our work, we are optimistic that our study will be enthusiastically received by the broad Nature Communications readership. We are also hopeful that this reviewer will find our revised manuscript, which incorporates additional key functional studies and a completely reworked text that improves readability, suitable for publication.

1. There is a large effort to mine publicly available and in-house generated RNA-seq and chemoproteomic datasets to make interesting observations and correlations. However, the majority of the manuscript is essentially mining one dataset after another to make correlations with very little follow-up on the functional relevance of identified sites. For example, the authors use their chemoproteogenomics approach to discover new variants (20 to be exact) that were not identified in COSMIC, CCLE or ClinVar. The highlight several of these variants in the results (HMGB1 R110C and SARS R302H) with hints towards function and move on to the next list of variants identified by chemoproteogenomics. The ligandability studies identified fragments that bind particular sites on protein variants (KB02 ligand for ALDOA Cys178 (G196G), HMGB1 Cys106 (R110C), HLA-B/C Cys125 (V127L/S123Y)), which sets up at the very least mutagenesis and biochemical binding assays and additional functional assays to elevate the impact of these findings. This is a recurring theme in the manuscript (providing a list of proteins/sites with little to no verification/follow up), which would be appropriate for a proteomics-focused journal but not for a broad audience.

We appreciate this reviewer's pointing out the value in follow-up studies to ensure the impact and validity of our findings. To this end, we first substantially re-organized and re-written our manuscript to improve the coherence and narrative and deemphasize the overtly technical data mining aspects. We also selected several targets for further analysis via biochemical assays. Specifically, we have found that the aforementioned HMGB1 R110C mutation affords a decrease in cytokine activity (now **Figure 6I,J** and copied below). Furthermore, using a tryptophan scanning approach, we show that bulkier modifications, which we expect to

mimic small molecule binding) afford a still more substantial decrease in cytokine activity. We also interrogated the impact of the CAND1 G1069C mutation, as CAND1 is known to be a key regulator of Cullin Ring E3 Ubiquitin Ligase activity. We find that this mutation completely blocks interactions between CUL1 and CAND1 and, quite unexpectedly, that the bulkier G1069W mutations rescue this interaction (now **Figure 6H** and copied below). Taken together these two examples highlight the capacity of our approach to detection variants of high functional significance. We have also reorganized that manuscript to improve accessibility to a broader range of reviewers, as we concur with this reviewer that the manuscript in its first iteration was overly technical and contained many details only accessible to proteomics specialists. Thus, we hope that this reviewer will concur that this added data together with our major rework of the text, have substantially improved our manuscript to a level meritorious of publication in Nature Communications, which is the home to many high impact proteomics studies.

New functional data now added to in Figure 6. H) HMGB1 proteins were tested for ability to induce TLR4-mediated immune response using HEK-Blue reporter cell lines (hTLR4 and Null control) and corresponding PRR assay. H) WT and G1069C mutant CAND1 proteins bind Cul1 while the G1069C CAND1 mutation perturbs binding. HEK293T cells were co-transfected with FLAG-Cul1 and the given HA-tagged CAND1 protein (WT, G1069C, or G1069W) or control FLAG-GFP. Anti-FLAG resin was used to pull-down FLAG-Cul1 from cell lysates along with any complexed proteins. Western Blots were incubated with the indicated primary antibodies, * indicates a non-specific HA band. I) Results show mean response ratios (error bars = SD, n = 4 per condition) of hTLR4 and Null cells to increasing concentrations ($\mu\text{g/mL}$) of WT, R110C, and R110W proteins as indicated over 2 independent experiments. AT = commercially-available all-thiol fully reduced HMGB1; diS = commercially-available di-sulfide HMGB1; working concentration of 0.2 $\mu\text{g/mL}$ for both. Reference lines on graph indicate EC_{max} (solid line), EC_{50} (dashed line), and H_2O control (dotted line) response ratios to canonical positive control ligand (LPS) specific to hTLR4 cell line. Significance determined via unpaired student's t-test; **= $p < 0.01$, ***= $p < 0.001$, ****= $p < 0.0001$. J) Response ratio curve of hTLR4 and Null cells to positive control ligand (LPS). EC values generated using nonlinear regression (Asymmetric (five parameter), X is concentration). Data provided in **Table S6**.

2. I also find the results section reads more like a methods section, which again goes back to point #1 above that the authors appear to be providing lists of protein, sites, variants, and correlations with existing and publicly available datasets with very little follow up. I appreciate the details on mass spectrometry method, but this level of methodological information makes it difficult to assess the biochemical and biological findings that I would think is the utility of this global approach. The results section ends with a methods like description of developing software to improve sensitivity of variant peptide identification. Again, the manuscript seems to be more appropriate for a proteomics-focused journal.

We apologize for the overly technical and somewhat disjointed nature of our initial manuscript submission. To address these concerns, which were also highlighted by the first reviewer, we have subjected the text to a major rework that includes, among other changes, substantially streamlining the genomics portion of the manuscript, reorganizing the results to begin with the proteomics method, addition of the aforementioned functional data, and substantially reworking both the main and supplementary figures in a manner that we believe improves the accessibility of our manuscript. We do acknowledge that there remain some unavoidable technical aspects to our manuscript, such as the highly significant two-stage FDR controlled search. We believe that this approach is a particularly important aspect of our manuscript—realizing the full potential of multi-omics on discovery of functional proteoforms, for basic and translational applications, will undoubtedly benefit from the improved rigor of our approach.

3. The cysteine profiling studies on HLA do not fit with the rest of the story presented. The majority of the manuscript has emphasized missense variants that result in an acquired cysteine in the context of cancer (KRAS G12C is used frequently as a model protein through the paper). The author continues this theme and perform ligandability screening of several cancer cell lines followed by a rather abrupt change to efforts for expanding HLA cysteine peptide coverage.

We thank this reviewer for pointing out this narrative inconsistency. While HLA have a clear connection to cancer vaccine and immunotherapeutics, we agree with this reviewer that the variants, as presented initially, did not fit with the cancer-focus of the rest of the manuscript. This and the other narrative incongruities prompted us to substantially revise our manuscript, as discussed above. We have included additional analyses of dbSNP and CLINVAR (e.g. **Figure 3E** **Figure 4G** and **Figure 5E**) to highlight the added value of including germline variants, such as those from the HLA, in proteogenomic analysis, both for improving detection of somatic variants, and for identifying interesting variant sites in proteins,

including for non-cancer directed applications. We have also modified the text to the following, which we believe adds value to our study as a resource of cell-line specific variant data for future efforts to characterize rare variants: *“Comparing the variants in our cell line panel to those found in dbSNP and ClinVar, we find that 25,735 (dbSNP/common) and 3,982 (ClinVar; 3,409 common and 573 rare) variants are found in our cell line panel, which highlights additional opportunities for analysis of acquired cysteines relevant to other genetic contexts, including rare disease and healthy genomes (Table S3).”* To incorporate this reviewer comment into our manuscript, we have also moved the HLA data to **Figure 4P**, where it fits much better as an example validation of variant reactivity from our proteomics data.

Additional comments:

1. The results section in general is very information rich. For example, the authors list metrics from chemoproteomics analysis (page 21 and 27). Are all these data necessary? I find the results section is difficult to read at times and this could be mitigated by reducing the sheer amount of details and focus on the interpretation and impact of the findings.

We thank this reviewer for pointing out key places where our manuscript is overly technical. While we believe that there is value in including all technical details in a manuscript to help facilitate rigor and reproducibility, we acknowledge that these sections together with several others detract from the readability of the manuscript. To address these points, we have removed these details to the supporting information in our revised submission. We have also added additional interpretation throughout the results section to improve readability. We believe that the analyses presented in the SI add value to the study and represent a general resource for interested readers, without overly complicating the message. However, if the editorial staff feels that additional material should be removed, we are happy to accommodate those requests.

2. Several of the figures are extremely dense and complicate (Figures 2, 4, 5, 6). Can these figures be simplified so that the main claims/findings are clear?

We apologize for the content-rich nature of our figures. We have made every effort to include workflows and to simplify graphics where possible. We do acknowledge that several of our figures are still quite information rich, which we feel helps to broaden the impact and depth of our study. We hope this reviewer will concur that these data rich figures add value. However, if they or the editorial staff have additional suggestions about specific content that can be removed, we are happy to consider additional modifications.

3. The supporting information has 40+ figures. Are all these necessary?

While we believe that the richness of our data is a key strength of our study, and one that was appreciated by both this reviewer and reviewer 1, we acknowledge that some of the supporting figures are likely superfluous to the central message. To this end, we have made every effort to condense our manuscript while maintaining the core essential findings. Of note, we have removed the previous **Figure S4, Figure S5, Figure S22**, and b.

REVIEWERS' COMMENTS

Reviewer #2 (Remarks to the Author):

I appreciate the authors addition of new functional data presented in Figure 6. While providing initial evidence, I don't find the selection of tryptophan mutations to mimic small molecule binding at gain-of-cysteine mutations convincing. Are the tryptophan point mutations specifically testing cysteine variant function or perhaps the tryptophan mutation is resulting in general misfolding of the protein? If the authors were to select sites not identified as gain-of-cysteine variants on CAND1 or HMGB1 would the resulting mutant proteins show comparable disruption of function? If so, it will be difficult to assign specific functionality to the cysteine variant using this approach without further mechanistic evaluation. The current data supports this observation. For CAND1, the G1069C mutation blocks protein-protein interactions with CUL1 while G1069W mutation rescues this effect. For HMGB1, both R110C and R110W mutants show a decreased TLR4 response. I believe the authors data support more complicated mechanisms with the cysteine variants that is difficult to generalize with a simple tryptophan mutation.

While the authors have substantially revised the manuscript text, I don't believe it is sufficient to address my previous major critique that this manuscript is more appropriate for a proteomics-focused analytical chemistry journal.

We thank this reviewer for their time and careful critique of our manuscript. While we understand their concerns about the relative scope of our study, we hope that they will concur with the other two reviewers that our manuscript does merit consideration for publication in Nature Communications.

Reviewer #3 (Remarks to the Author):

This manuscript describes a chemoproteogenomic workflow to identify gain-of-cysteine mutations and mutations proximal to reference cysteines within a panel of cell lines with varying levels of missense mutation burden. The workflow utilizes RNA-Seq data to generate cell line-specific variant databases that are then matched in a 2-stage sequence to identify peptides with acquired cysteines, or mutations proximal to known cysteines. This workflow was applied to a panel of 11 cell lines with varying levels of mutation burden. RNA and exome sequencing as well as chemoproteomics were applied in this panel of cell lines to identify 59 gained cysteines and 302 cysteine-proximal mutations. Ligandability screening was performed in 3 cell lines using a small panel of electrophilic fragments, identifying 27 liganded cysteines. Two liganding sites were further characterized using C to W mutations to mimic a liganding event. Introduction of C to CAND1 blocked interaction with CUL1, and this was reversed by the W mutation. Introduction of C and W to HMGB1 resulted in reduced response to TLR4.

The authors have addressed many of the comments in the prior round of review, and have restructured their manuscript to improve the general readability. Importantly, they have now included follow-up characterization on the liganding of CAND1 and HMGB1, which is important to demonstrate the functional relevance of the cysteine mutations, and addresses some of the concerns of reviewer 2 regarding the proteomics-only nature of the original submission. Given the general responsiveness to prior comments, I recommend publication of this manuscript in Nature Communications in its current form.

We thank this reviewer for their positive endorsement of our study as meritorious of publication in Nature Communications in its current form.

Reviewer #4 (Remarks to the Author):

In their manuscript, Desai et al. expand their chemoproteomics platform to include missense variant Cys residues in addition to canonical Cys. Though certain high-value Cys variants are well known (e.g. KRAS G12C), it turns out that such variants are relatively common in at least some cancers (and cell lines) and could present promising therapeutic targets, though until now they've been invisible to chemoproteomics screens. Here the authors first describe some changes they've made to MSFragger/FragPipe (software for analyzing MS data) to identify these variant peptides. They then go over a series of bioinformatics analyses they undertook to understand the prevalence of missense variants and to select cell lines with high levels of missense variants suitable for targeting missense Cys (and other missense variants near canonical reactive Cys) in their screens. Next, they use chemoproteomics to identify such sites and perform additional informatic analyses to examine the factors that influence which sites they detect. Along the way, they follow up on a couple of specific examples.

Overall, I think this manuscript addresses an important and interesting problem. The work they present is well done and enhances our understanding of these Cys missense variants. Though I didn't see the first draft of this manuscript, it's apparent from the reviews and rebuttal that it underwent extensive revisions. I think the revised version is clear and well written and does a good job of balancing technical details with their broader biological/chemical message.

We thank this reviewer for their positive assessment of our study and their comprehensive critique and exciting ideas for follow-up studies.

The editor asked me specifically to comment on the points raised by Reviewer #2. It seems to me that the revisions made by the authors have gone a long way toward addressing these issues. I can see where the reviewer's concern about the technical nature of the work is coming from, for example given the discussion of database searching strategies. Having said that, I think those details are essential to the work, and in my opinion the authors do a good job in this revision of describing that work in a succinct and accessible way. Another concern raised by Reviewer #2 was that there isn't enough follow-up and the various bioinformatic/chemoproteomic analyses aren't well integrated. In its current presentation, I think the chemoproteomics data are clearly presented while the data mining efforts provide important context. And the follow-up work targeting HMGB1 and CAND1 is a significant enhancement. Overall, I do think this paper warrants publication in Nature Communications. Chemoproteomics is inherently interdisciplinary, and I think the authors' work integrating chemoproteomics with proteogenomics to enable drug discovery is conceptually novel enough to warrant presentation to a broad audience – not just proteomics specialists.

One additional comment: the key question in this paper is why the authors detect few missense variants in their chemoproteomics datasets, relative to the number they detect via sequencing. The authors already consider a variety of factors, but I think there's at least one more worth considering. Another biologically intriguing possibility is that these missense mutations are hard to detect in protein form because they tend to make the proteins less stable and prone to degradation, and thus don't persist as long as their wildtype counterparts. The comparison the

authors do with overall protein abundance levels doesn't fully account for this possibility because each mutation will only affect one copy of each gene, and the cell lines likely have another normal copy and express a mixture of mutant and wildtype protein. In cases where the mutated form is degraded preferentially, there may still be enough copies of the wildtype form around to maintain overall protein levels, or at least minimize their change. There are already some hints in the data that this could be going on. For example, in line 367 the authors note that "nearly all of the 77 variants in Clinvar and identified by chemoproteomics are annotated as benign". One explanation for this could be that deleterious variants tend to disrupt protein function in ways that lead to their premature degradation and prevent their detection. Along these lines, around line 360 you compare the frequency of deleterious mutations among various subsets of your chemoproteomics data. I'd be curious to know how the frequency of predicted deleterious mutations compares between the SAAV's you detect and the ones you do not detect. Experimentally, you could even test this by looking at mutant peptide abundances (or comparing mutant and non-mutant sequence pairs) following proteasome inhibition, though that may be beyond the scope of this paper.

This is a wonderful suggestion and one that we are excited about pursuing further in follow-up studies. Indeed, a key goal of this study and particularly our rigorous two-stage search strategy was to set the stage for a follow-up study that tackles variant-induced changes to protein half-life. We did do an analysis looking at the CADD scores of detected versus undetected variants, and, while underpowered, this analysis did not show substantial differences in CADD scores between these two categories of variants. Given the small sample size and potential limitations of CADD in stratifying the deleteriousness of missense variants, we opted to proceed conservatively and omit this data from our current manuscript. As the reviewer alludes to there are very intriguing time-dependent labeling experiments, including using proteasome inhibition, to delineate variant-induced changes to protein stability. We look forward to pursuing these studies, using our platform in followup studies. To address this point and vision, we have modified the discussion to the following: "Furthermore, we expect that a subset of the tough-to-detect variants may cause decreased protein stability and premature degradation. Such variant-induced changes in protein half-life may be detectable by pairing proteasome inhibition with chemoproteogenomic analysis."

Another issue the authors consider is the effects of variants on cys oxidation. Around line 435 the authors look at effects of missense mutations on nearby reactive Cys residues and note that they can alter oxidation state in some cases. A related potential reason why they don't detect many missense mutations that introduce new Cys residues could be that while evolution tends to place reactive Cys in chemical environments that favor maintaining their reactive state, these missense Cys arise sporadically and are more likely to occupy comparatively inhospitable environments, leaving them prone to oxidation or otherwise unreactive and thus not detectable via chemoproteomics. This probably isn't straightforward to address experimentally, but is worth mentioning in the text.

We are delighted that this reviewer finds value in our analysis in understanding the cross-talk between variants and cysteine oxidation

Reviewer #4 (Remarks on code availability):

I didn't look at the code in detail because I was focusing my attention on the issues raised by Reviewer #2. But I can look at it more closely if needed.